# Pharmacokinetics of Antibacterial Agents in the Elderly: The Body of Evidence

**DOI:** 10.3390/biomedicines11061633

**Published:** 2023-06-04

**Authors:** Olga I. Butranova, Elena A. Ushkalova, Sergey K. Zyryanov, Mikhail S. Chenkurov, Elena A. Baybulatova

**Affiliations:** 1Department of General and Clinical Pharmacology, Peoples’ Friendship University of Russia named after Patrice Lumumba (RUDN University), 6 Miklukho-Maklaya St., 117198 Moscow, Russia; ushkalova-ea@rudn.ru (E.A.U.); sergey.k.zyryanov@gmail.com (S.K.Z.); chenkurov-ms@rudn.ru (M.S.C.); baybulatova-ea@rudn.ru (E.A.B.); 2State Budgetary Institution of Healthcare of the City of Moscow “City Clinical Hospital No. 24 of the Moscow City Health Department”, Pistzovaya Srt. 10, 127015 Moscow, Russia

**Keywords:** elderly, pharmacokinetics, absorption, distribution, metabolism, excretion, antibiotics

## Abstract

Infections are important factors contributing to the morbidity and mortality among elderly patients. High rates of consumption of antimicrobial agents by the elderly may result in increased risk of toxic reactions, deteriorating functions of various organs and systems and leading to the prolongation of hospital stay, admission to the intensive care unit, disability, and lethal outcome. Both safety and efficacy of antibiotics are determined by the values of their plasma concentrations, widely affected by physiologic and pathologic age-related changes specific for the elderly population. Drug absorption, distribution, metabolism, and excretion are altered in different extents depending on functional and morphological changes in the cardiovascular system, gastrointestinal tract, liver, and kidneys. Water and fat content, skeletal muscle mass, nutritional status, use of concomitant drugs are other determinants of pharmacokinetics changes observed in the elderly. The choice of a proper dosing regimen is essential to provide effective and safe antibiotic therapy in terms of attainment of certain pharmacodynamic targets. The objective of this review is to perform a structure of evidence on the age-related changes contributing to the alteration of pharmacokinetic parameters in the elderly.

## 1. Introduction

The global population aging in the 21st century is unprecedented. In the Western world persons over 65 years are the fastest growing cohort [1], which outnumbers the population of children below five years old and attracts the attention of researchers all over the world [2]. It is predicted that by 2100 in Europe people over 65 will make up 31% of the total population, and people over 80 will reach about 15% [3].

Age-related physiological and pathological changes, poor functional status, poor nutrition, and comorbidities predispose older adults to infections and their complications [4,5]. The incidence and severity of infections increase with advancing age [6,7]. Compared to younger age groups, elderly patients are more prone to pneumonia, skin and soft tissue infections, urinary tract infections and septicemia [1,4]. An additional problem is a substantial risk of antibiotic (AB) resistance; its typical risk factors in the elderly include frequent contact with the healthcare system, frequent AB exposure, depressed immune system, frailty, and comorbidity [4]. Elderly patients are considered the high-risk group for the development of healthcare-associated infections caused by multidrug-resistant (MDR) bacteria [8,9,10]. The elderly population has longer hospital stays compared to younger adults and a significantly higher mortality rate (25%) compared to the general population (10%) [6,11,12]. Infections aggravate the course of concomitant chronic diseases, including cardiovascular and cognitive disorders, and contribute to the emergence of a new comorbidity [13].

High infectious morbidity leads to high consumption of antimicrobial agents by the elderly. ABs are among the most frequently prescribed medicines to seniors [14,15,16] and their use is accompanied by a significant rate of side effects and clinically relevant drug-drug interactions compared to younger counterparts [14]. Adverse drug reactions (ADRs) are an important cause of morbidity and mortality in the elderly [17,18,19] and their risk is significantly increased in the presence of comorbidity and polypharmacy [19,20,21,22].

The choice of optimal antimicrobial agent for the elderly is challenging [23]. Finding the right balance between efficacy, safety and tolerability of antibiotics is difficult for several reasons including significant changes in body tissue composition, a progressive physiological decline of organ functions, frailty, comorbidity, and polypharmacy [24]. All these factors can cause significant alterations in antimicrobials pharmacokinetics (PK) and pharmacodynamics (PD) leading to altered efficacy, safety, and tolerance. The problem is compounded by the fact that elderly patients represent a heterogeneous group that should be treated individually [25,26].

This review includes an analysis of the available data on the PK of antibacterial agents in the elderly and a consideration of the critical issues of AB use in this vulnerable heterogenous population. We used the PubMed database to retrieve relevant articles dedicated to the pharmacokinetic studies of antibacterial agents in the elderly published during the period 980–2023 years.

## 2. Factors Influencing AB Prescribing in the Elderly

Infections in the elderly may be caused by a more diverse group of pathogens compared to the younger population [6,27,28]. For example, there is a higher prevalence of Gram-negative bacilli in pneumonia and a lower prevalence of *E. coli* in urinary tract infections [6].

Common infections do not manifest with classic symptoms in the elderly. Aged patients may have neither fever nor leukocytosis [29]. An absence of fever and a lack of respiratory symptoms has been described in 40–60% of elderly patients with community-acquired pneumonia [30]. The only clinical presentation of pneumonia in up to 20–50% of the elderly may be an altered mental status including delirium and confusion, a sudden decline in functional capacity, and worsening of underlying diseases [28]. The high prevalence of unusual and/or multidrug-resistant pathogens in the elderly makes AB susceptibility testing highly desirable, though in the real clinical practice, antibiotics may be prescribed empirically as even subtle clinical manifestations may herald the onset of life-threatening infectious disease and delayed therapy can worsen treatment outcomes [28].

Appropriate AB dosing requires knowledge of the pharmacokinetic and pharmacodynamic properties of AB which are often altered due to aging processes, including age- or disease-related decline of kidney and liver functions. To select an optimal AB for the concrete patient, it is necessary to identify all comorbidities, concomitant drugs, and dietary supplements, and collect the patient’s allergic history. Consideration should be given to the factors associated with poor treatment compliance such as poor vision and/or hearing, physical dexterity, cognitive impairment, or mental illness [21,31,32,33]. These patients require treatment supervision by relatives or caregivers.

Elderly patients are at high risk of potential harm associated both with missed treatment and excessive AB therapy [31,34]. In long-term care facilities 50–75% of residents receive at least one course of AB each year with 30–50% of AB prescriptions being unnecessary or inappropriate in terms of drug choice, dosing regimen and/or duration of treatment [35]. Inappropriate drug selection and use may lead to medication-related problems, including ADRs, therapy failure and withdrawal events alongside the spread of AB resistance [34,36].

Compared to younger counterparts, older patients are more vulnerable to AB side effects and clinically relevant consequences of drug interactions [14]. The risk of ADR development is especially high in patients with comorbidity and polypharmacy [19,20,21,22]. ADRs are an important cause of morbidity and mortality in elderly patients [17,18,19].

Increased risk of AB-induced toxicity, ADRs, and negative outcomes in elderly patients with infectious diseases may be mediated by the changed PK of AB resulting in the changed PD.

## 3. General Considerations on AB Pharmacokinetics in the Elderly

The development of knowledge of the antimicrobial PK/PD relationship is essential to provide maximization of the efficacy, minimization of the toxicity, and preservation of the lifespan of currently available antibiotics [37].

PK characterizes the concentration time course of antibiotics based on absorption, distribution, metabolism, and elimination [38]. The value of plasma concentration being the result of PK processes is the main determinant of PD. PD characterizes parameters of antibacterial activity and describes the effect of AB on the target pathogens, relying on the minimum inhibitory concentration (MIC) [38,39,40].

The quantitative relationship between PK parameters and PD parameters is described by pharmacokinetic/pharmacodynamic (PK/PD) indices [41]. These indices should be used in patients with critical illness, central nervous system infections, severe burns, severe impairment of renal function, severe hypoalbuminemia, morbid obesity, and other medically complicated conditions. AB efficacy is described by the next PK/PD indices: the ratio of the area under the concentration-time curve (*AUC*) from zero to 24 h (*AUC0–24*) to the MIC, the ratio of the maximum plasma concentration (*Cmax*) to the MIC and % of the time during which the free plasma concentration exceeds the MIC *(% fT* > MIC) [42].

Based on PK/PD indices antibacterials can be divided into 3 groups [38]:time-dependent (β-lactams, natural macrolides, lincosamides, oxazolidinones),concentration-dependent (aminoglycosides, fluoroquinolones, nitroimidazoles, daptomycin, quinupristin/dalfopristin),concentration-dependent with time-dependence (tetracyclines, glycylcyclines, glycopeptides, semisynthetic macrolides).

The efficacy of time-dependent antibiotics is mainly related to *%fT* > MIC. A significant increase in concentration does not enhance the antibacterial effect of these antibiotics, hence dosing regimens maintaining stable drug concentrations above the MIC are preferred [43].

The efficacy of concentration-dependent antibiotics is defined by *Cmax*/MIC ratio. Antibacterial activity of this group increases with increasing concentration of AB; therefore, treatment success is determined by a larger dose of AB with less frequency of administration [43].

The efficacy of concentration-dependent drugs with time-dependence is determined mainly by the ratio *AUC0-24*/MIC [44]. The aim of treatment with this type of antibiotic is to maximize the patient’s overall exposure to the drug [44].

The PK of antibiotics in the elderly may significantly differ from that in younger adults. Age-associated PK changes strongly depend on the patient’s individual characteristics, including individual rate of senescence of organs and tissues, comorbidity profile, presence, and rate of progression of geriatric syndromes (frailty is one of the most important), and severity of the main disease. The main contributors to altered pharmacokinetics in the elderly are age-related changes in organ mass and blood circulation alongside changes in body composition, and disease-associated changes in the organs and systems functioning.

### 3.1. Absorption

Drug absorption is an essential process of drug transport from the site of administration to the systemic blood flow. Depending on the route of administration, the rate of absorption may vary, resulting in a certain value of bioavailability—the fraction of unmetabolized drug that reaches systemic circulation. The main PK parameters describing absorption are *AUC* and *Cmax*. Decreased absorption leads to the decreased concentration of AB, and thus, to the decrease of treatment efficacy. On the contrary, increased absorption may result in increased concentrations and, thus, increased risks of toxic effects.

The most convenient route of drug administration is oral. In the elderly, there are both structural and functional changes of the gastrointestinal tract (GIT) which may affect absorption, and, thus, bioavailability of AB. Senescence results in the decrease of salivary glands secretion and changed the quality of saliva, atrophic changes in the GIT mucosa, damage of enteric neurons, gastric secretion, and GIT motility [45]. Elderly patients are characterized by delayed gastric emptying, reduced GIT blood flow, and alterations in pH, typically hypochlorhydria [8]. Since elderly patients are characterized by the presence of different chronic diseases and a high burden of polypharmacy, these factors should also be considered together with the physiological changes. Age-associated GIT alterations may be worsened in the presence of some comorbidities, drug interactions and ADRs. Senescence-related changes and drug-induced changes of GIT contributing to the PK changes are demonstrated in Table 1.

Age-associated and PPI-induced hypochlorhydria may result in the decreased absorption, and, thus, bioavailability of such antibacterials, as azithromycin, erythromycin, cefaclor, ceftibuten, sulfonamides [8], decreased intestinal motility and blood flow may lead to the decreased bioavailability of cefpodoxime proxetil [8]. The reduced first-pass metabolism in the elderly may lead to some decrease in active moiety formation from the prodrug form. Among antibacterials, there are several prodrugs. Ampicillin prodrugs include pivampicillin, talampicillin bacampicillin, and hetacillin (all are esters), another is sultamicillin (ampicillin linked to sulbactam by a methylene group) [77]. Examples of prodrugs among cephalosporines include ceftaroline fosamil and a combined form, novel cephalosporin-fluoroquinolone prodrug, including ciprofloxacin linked to the cephem core via the carboxylic acid [78]. An old antibacterial agent in the prodrug form is metronidazole, which is activated through reduction with redox active form production. The effect of the changed intestinal permeability on the absorption of antibacterials may be well illustrated with an example of inflammatory bowel disease, which incidence is rising in the elderly [79]. For patients with inflammatory bowel disease, a decrease of metronidazole exposure after oral intake was demonstrated as well as alterations of bioavailability of many other drugs administered orally [80].

Absorption is also affected by the number and function of transport proteins in the GIT. One of the most important transport proteins, P-glycoprotein (P-gp) revealed nearly no change in the elderly compared with younger adults, though intestinal P-gp activity was significantly reduced in the elderly with renal failure [81]. The same change was demonstrated for the organic anion transporter polypeptides 1B1 (OATP1B1), those activity was reduced in the elderly with chronic kidney disease (CKD) compared to healthy young participants and healthy elderly patients [81]. The activity of another intestinal efflux transporter, breast cancer resistant protein (BCRP), revealed a marked decrease both in the healthy elderly and those with CKD [81]. Estimation of BCRP expression in the liver demonstrated the same decline with age, being the lowest in the elderly compared with adults and children [82]. Aging also may also affect the expression of peptide transporters 1 and 2 (PepT1 and 2), which are involved in the uptake of di- or tripeptide substrates in such locations, as the intestine, kidneys, bile duct epithelium (PepT1), brain, lung, and mammary gland (PepT2) [83,84]. PepT1 and Pept2 expression may be altered in the presence of age-related diseases including diabetes mellitus (downregulation of PepT1) and obesity (leptin-dependent activation of PepT1 activity and expression) [83].

Some age-associated changes of transport proteins may contribute to the altered PK of AB, though genetic polymorphism is considered to be a more important factor resulting in these changes [85]. The only transport protein whose expression was found to be age-dependent was revealed with a quantitative review of age- vs. genotype-related differences, including P-gp [85]. Different results were demonstrated with liquid chromatography tandem mass spectroscopy transport proteins quantification: no age correlation was estimated for the hepatic protein expression of OATP1B1, OATP1B3, OATP2B1, or P-gp (*p* < 0.05) [86]. In general, there are limited studies aimed at the estimation of age-associated transport protein changes in humans. Animal studies declare a decline of gene expression and mRNA expression of most OATPs with aging (OATP1A1 OATP1A2, OATP1A4, OATP1A5 and OATP1B2), as well as of P-gp and multidrug-resistance like proteins 1 and 2 (MRP1, MRP2) [87,88,89].

The summary of age-associated changes of the main transport proteins involved in AB transport along with data on antibiotic-substrates is given in Table 2.

Absorption of orally administered antibiotics may be affected by food intake. For highly lipophilic drugs absorption is increased in the fed state. Food also changes acidity values in different departments of GIT, affecting the absorption of weak acids and bases. Passive diffusion and thus a high rate of absorption is specific for the unionized forms, and their formation depends upon the dissociation constant of the drug in the physiological pH range [108]. Fed state results in the increase of acidity in the stomach and, in less extent, in the colon. On the opposite, in the duodenum, jejunum and ileum fed state lead to the decrease of acidity. With pH greater than 5 ionization rate of weak bases is dramatically decreased, while for weak acids it is significantly increased [109]. Depending on the drug, taking on an empty stomach (preprandial) or with a meal (postprandial) may be recommended. In the elderly, for whom cognitive disorders are specific, as well as multiple comorbidities and polypharmacy, the risk of not meeting these recommendations is high, resulting in the decreased efficacy of treatment or increased risks of toxic reactions [109]. Age-associated changes also may aggravate fast-fed variabilities. Choosing antibiotics with high absorption regardless of food is strongly recommended for the elderly. Table 3 contains information about food effects on antibiotics absorption and related PK parameters along with recommendations on the proper administration.

Drug absorption in non-oral routes of administration (injection, inhalation) may also be affected by age-associated changes. Considering intramuscular drug administration skeletal muscle mass decrease and hypoperfusion are the main factor altering absorption. For intranasal administration and for inhalation the decreased blood flow is also a key determinant resulting in the reduced absorption. Table 4 includes data on the age-associated changes of absorption for non-oral routes of drug administration in the elderly.

### 3.2. Distribution

Drug distribution through different body fluids, organs, and systems is markedly affected by the aging process resulting in the change of such PK parameters, as the volume of distribution (*Vd*). The main contributors to *Vd* alterations are changes in the cardiovascular system specifically for the elderly population. Cardiovascular senescence includes the cascade of physiological changes which may precipitate the formation of cardiovascular diseases. Advanced age results in the thickening of the walls of arteries with a decrease in vessel compliance and an increase in pulse wave velocity resulting in systolic blood pressure increase. These changes lead to the left ventricular afterload contributing to myocardial remodeling and congestive heart failure (HF) as well as to ischemic heart disease [132]. HF mediates hypoperfusion first in the peripheral tissues, but with HF progression blood flow will be reduced in the GIT, liver and kidneys resulting in significant PK changes [128]. Oedema secondary to HF, ascites secondary to cirrhosis, and chronic liver disease—all can worsen a fluid accumulation in the site of infection and adjacent tissues. This leads to the dilution of AB concentrations at the infection site and predisposes to a treatment failure [133]. Systemic inflammation in the elderly can promote endothelial dysfunction, leading to increased capillary permeability and plasma leakage into the interstitial space, which in critically ill patients can be further aggravated by fluid administration. Consequently, *Vd* especially of hydrophilic antibiotics (e.g., beta-lactams) can increase and a loading dose may be required [134].

Another concern is a change in the body’s water and fat composition. In the elderly, there is an increase in the body’s total fat mass, fat redistribution (increase in the abdominal fat and reduction in subcutaneous fat), and fat infiltration of various organs (liver, pancreas) [135]. Total body water content is decreased in the elderly and obese patients, and intracellular water is also reduced reflecting skeletal muscle loss [136]. An increase in fat mass may affect drug distribution, since the presence of obesity mediates changes in the tissue blood flow and perfusion. Considering the distribution of ABs, tissue penetration may be altered, as shown in the studies with cefuroxime and ciprofloxacin in obese patients [137].

Physicochemical drug characteristics are other determinants affecting *Vd*. Lipophilic agents can easily cross cellular membranes and blood-tissue barriers distributing to the various organs and tissues and resulting in the high values of *Vd*, while hydrophilic drugs are mainly concentrated in the body fluids and are typically characterized by the low values of *Vd*. Lipophilic antibacterial agents include fluoroquinolones, macrolides, tetracyclines and glycylcyclines, oxazolidinones, rifampicin, and chloramphenicol. Hydrophilic antibiotics are β-lactams, aminoglycosides, and glycopeptides. With increased fat content *Vd* for highly lipophilic drugs increases. Advanced age is associated with the decrease of total body water resulting in *Vd* contraction of hydrophilic drugs (e.g., aminoglycosides, β-lactams, and glycopeptides), though in critically ill patients *Vd* of hydrophilic agents may be increased with hemodynamic insufficiency development, excessive permeability of capillaries, and ongoing infusion therapy [138]. The ability of both lipophilic and hydrophilic antibiotics to cross blood-tissue barriers may be increased in the presence of some diseases. The most important barrier is the blood-brain barrier, and its breakdown may result in an increased risk of drug-induced neurotoxicity. The elderly population is characterized by a high prevalence of pathological states resulting in increased permeability of the blood-brain barrier, such as Alzheimer’s disease, Parkinson’s disease, Huntington’s disease, amyotrophic lateral sclerosis, multiple sclerosis, and ischemic stroke [139]. AB therapy in patients with listed diseases may be accompanied by the rise of various neurotoxic reactions. The highest risk of encephalopathy was reported for penicillins, cephalosporines, carbapenems, oxazolidinones, fluoroquinolones, polymyxins, sulfonamides, metronidazole. Tetracyclines can induce cranial nerve toxicity and intracranial hypertension [140].

Another factor affecting drug distribution is plasma protein binding rate (PPB). Serum albumin, lipoproteins and alpha-1 acid glycoprotein are the main plasma proteins sequestering drugs in plasma [141]. A bound fraction of a drug is pharmacologically inactive and stays in the intravascular space, while an unbound fraction penetrates the extravascular space, reaches corresponding molecular targets, and causes a pharmacological response. Changes in the quantity and quality of plasma proteins may result in a disbalance between the bound and unbound fractions leading to altered pharmacological effects. Age is one of many factors that can influence drug protein binding, but the clinical significance of age-related hypoalbuminemia seems to be minimal. In healthy noninstitutionalized individuals, a gradual small decrease in serum albumin level (approximately 4% per decade) was found [142]. It is not noticeable until people reach 70 years of age [142], when the level of serum albumins decreases by 20% [143].

Hypoalbuminemia is more specific for elderly hospitalized patients. Serum albumins decrease below 3.5 mg/dL was observed with aging, and among cases of marked hypoalbuminemia at hospital discharge (<2.5 mg/dL) 74.2% were reported in persons over 65 years of age [144]. Serum globulins are also affected by aging, in 47.6% of patients aged between 60 and 85 years an increased gamma gap was observed (>3.1 g/dL) [145]. More pronounced alterations of serum proteins may be caused by acute or chronic disease, proteinuria, and malnutrition [142].

Hypoalbuminemia may play an important role in critically ill patients treated with intravenous highly protein-bound antimicrobials such as cefazolin, ceftriaxone, ertapenem, sulfonamides, clindamycin and daptomycin [146]. Decreased PPB may result in a significant increase in free serum concentrations of these antibiotics [146,147] which may require direct measurement of free drug levels [147]. In general, hypoalbuminemia may lead to the increase of Vd of highly albumin-bound antibiotics and hydrophilic drugs (e.g., streptomycin) and to the decrease of Vd of the α1-acid glycoprotein-bound drugs (e.g., rifabutin) [43]. In a retrospective observational study, older patients with methicillin-resistant Staphylococcus aureus hospital-acquired pneumonia and severe hypoalbuminemia had significantly longer vancomycin half-life (T1/2), high values of AUC, more frequent nephrotoxicity episodes, and greater risk of 28-day mortality compared with patients with mild hypoalbuminemia [148]. The authors recommended individual vancomycin dose adjustment to senior patients with low body weight and severe hypoalbuminemia. In the presence of hypoalbuminemia highly protein-bound ertapenem (normal PPB is 85–95%) showed a higher incidence of 30-day mortality in patients with carbapenems-susceptible Enterobacteriaceae compared to less protein-bound imipenem or meropenem [149].

Other studies demonstrated a decreased probability of target attainment with ceftriaxone in critically ill patients with severe hypoalbuminemia [150], and an increased risk of clinical failure with this AB [151].

Older individuals have increased plasma levels of α1-acid glycoprotein that are associated with a reduced unbound fraction of basic antibiotics, e.g., macrolides [8]. The synthesis of α1-acid glycoprotein may be augmented by infections and malnutrition [43] and an increase of its levels from 2 to 6-fold is seen in severe inflammation and cancer [141], which can affect Vd of antibiotics binding mainly to alpha-1-acid glycoprotein, such as clindamycin [152].

Nutritional status has a significant impact on antibiotics distribution and to a lesser extent on other PK parameters. Both obesity and malnutrition (undernutrition) are highly prevalent among older adults [153]. Rates of obesity vary in different age groups and are the highest among the young-old individuals (65–74 years) [154]. Obesity (BMI > 30 kg/m^2^) and particularly morbid obesity (BMI > 40 kg/m^2^) influence various physiological processes including gut permeability, gastric emptying, cardiac output, liver and renal function, and is associated with the different physiological compositions of muscle and fat compared to non-obese patients [137,155]. An increase of both body fat tissue and lean body mass in patients with obesity leads to an increase of Vd particularly of lipophilic drugs. The elimination of highly lipophilic agents, such as fluoroquinolones, macrolides, oxazolidinones, tetracyclines and rifampin can decrease. Morbid obesity (BMI > 40 kg/m^2^) can profoundly affect both antibiotics distribution and clearance [156]. PK changes in obese patients can potentially reduce the efficacy of standard AB doses used for the treatment of non-obese individuals [157]. In general, the dosing of lipophilic antibiotics in obese patients is recommended to be based on actual body weight, and that of hydrophilic antibiotics—on ideal body weight [158], but optimal dosing in obese elderly patients needs further study.

Malnutrition is another important concern in the elderly [159]. In malnourished individuals, there is a decrease in adipose tissue content and lean body mass with an increase in total body water. Malnutrition is associated with other pathophysiological changes which can impact PK such as hypochlorhydria, delayed gastrointestinal emptying time, increased, or decreased intestinal transit time, gastric and mucosal atrophy and dysfunction, gastrointestinal inflammation, and pancreatic insufficiency [43]. P-gp activity in the enterocytes of malnourished patients is decreased and tight junctions are enlarged, influencing the uptake of food and drugs [43]. A common feature of malnutrition especially in elderly patients with infections is pronounced hypoalbuminemia [7,43], those effects on *Vd* were discussed above.

The list of factors affecting *Vd* and clearance in the elderly with severe infections and their interrelationships are shown in Figure 1.

### 3.3. Metabolism

Age is associated with significant changes in drug metabolism. In healthy aging the mass of the liver reduces by 20–40% resulting in a reduction of drug clearance [31]. A decrease in liver mass and liver function is mainly related to the significant hepatic blood flow decline (40 to 60%) in the elderly [24]. Both decreased hepatic function and reduced hepatic blood flow contribute to increased *T1/2* of hepatically metabolized antibiotics in the elderly [8]. The age-related loss of surfaced endoplasmic reticulum causes a strong negative correlation between age and hepatic microsomal phase I drug metabolizing activity [160]. In people aged ≥70 years activity of the cytochrome P450 (CYP450) oxidases may decrease by 30% [161] resulting in the decline of clearance of CYP substrates. CYP3A activity decrease was reported in the elderly compared with healthy adults, resulting in the midazolam and atorvastatin Cmax nearly twice increase [81]. In the elderly 30 to 50% clearance reduction was reported for the CYP3A4 metabolized drugs [24] and 20% reduction for CYP2D6 substrates [23].

Changes in the body composition specific to the elderly may affect the functions of drug metabolizing enzymes. Kaburaki S et al. (2022) observed associations between the skeletal muscle mass index (SMI), handgrip strength (HGS), hepatic steatosis index, and activity of CYP2C19 and CYP3A4. In male patients ≥65 years of age a reduction in SMI and HGS below the sarcopenia diagnostic criteria correlated with a decline in CYP2C19 and CYP3A4 activity. In elderly female patients, a decline in CYP2C19 metabolic activity was associated with fatty liver disease presence [162].

Liver pathology being highly prevalent in the elderly population is the most obvious factor altering CYP450 expression and activity. In this respect, it is interesting to consider the results of the estimation of the protein abundance and gene expression of various CYPs in the liver samples of patients with hepatitis C, alcoholic liver disease, autoimmune hepatitis, primary biliary cholangitis and primary sclerosing cholangitis. CYP2E1 was defined as the most vulnerable enzyme in which protein levels were significantly reduced in Child–Pugh class A cirrhosis. The most prominent downregulation of metabolizing enzymes was associated with alcoholic liver disease (CYP1A2, CYP2C8, CYP2D6, CYP2E1, CYP3A4, UGT2B7) and primary biliary cholangitis (CYP1A1, CYP2B6, CYP2C8, CYP2E1, CYP3A4). The protein abundance most of UDP-glucuronosyltransferases (UGT) was unaffected by liver pathology (UGT1A1, UGT1A3, UGT2B15) [163].

Decreased function of CYP450 enzymes may lead to the reduction of the first-pass metabolism of orally taken macrolides, fluoroquinolones (except levofloxacin), clindamycin, tetracyclines, sulfamethoxazole/trimethoprim, and rifampin, resulting in the increase of bioavailability and serum concentrations of these agents [8]. Though these changes in enzymes metabolizing activity vary significantly from drug to drug and from person to person they might be an important cause of ADRs [160].

The activity of the phase II enzymes, such as sulfotransferases, UDP-glucuronosyltransferases (UGTs), and glutathione s-transferases (GSTs) is usually less affected by age and therefore, the clearance of moderately lipophilic antibiotics, such as fluoroquinolones or linezolid is similar to that among young adults [24]. There is some evidence that phase II metabolism might be affected by frailty [24].

Malnutrition can cause a decrease in the content of hepatic cytochromes, which was proved by the observed reduction of drug metabolism in patients with cachexia [164].

Substrates, inhibitors, and inducers of CYP450 isoenzymes among antibacterial agents and CYP450 aging changes are indicated in Table 5.

### 3.4. Excretion

Elderly patients have a high risk of decreased antibiotics clearance from the body due to declining functions of the lung, bladder, liver, GIT and circulatory system, but deterioration of the kidney function is the most important [207]. The kidneys are the major route of elimination for many classes of antibiotics including beta-lactams, aminoglycosides, glycopeptides, fluoroquinolones (except moxifloxacin), lipoglycopeptides, lipopeptides (daptomycin), trimethoprim/sulfamethoxazole [8].

A gradual decrease in the kidney size and weight, renal blood flow, estimated glomerular filtration rate (eGFR), altered renal tubular secretion and age-related anatomic abnormalities (e.g., glomerulosclerosis, arteriosclerosis, arteriolar hyalinosis, medial hypertrophy, tubular atrophy) leads to a progressive decline of the renal function in the elderly [208]. Renal mass reaches about 400 g in the fourth decade of life and declines gradually to about 300 g [209].

Both age-related physiological changes and pathological changes (due to hypertension, diabetes mellitus, and heart failure) lead to the amplification of the cellular signaling pathways involved in renal cell senescence resulting in the imbalance between the proliferation and apoptosis with the intensification of the last one [210]. Renal aging results in increased susceptibility to acute kidney injury (AKI) and increased risk of formation of chronic kidney disease (CKD). In the age group 40 to 49 years CKD stage 3–5 was reported only in 1.4%, in the group 50 to 59 years—in 5.4%, while for the group 70 to 79 years—in 35.4%, and in the group 80 to 89 years—in 30.9% [211]. By the eighth decade of life approximately 30–40% of all glomeruli become sclerotic and by the ninth decade kidney size and the total number of glomeruli may be about 70% of that of the third decade [212]. Patients ≥80 years have a 40–50% decline in renal function compared to adults of middle age [1,213]. The decline in renal function is accelerated in patients with frailty [214].

Another factor contributing to the altered renal function and AKI is the use of nephrotoxic drugs. Antibiotics may result in nephrotoxic reactions with various mechanisms, including acute tubular necrosis and acute interstitial nephritis (Table 6). In the elderly decreased renal clearance mediates the amplification of nephrotoxicity, since for the majority of associated antibiotics it has a concentration-dependent character.

AKI risk in patients using antibiotics is increased with increasing age: odds ratio (OR) was 4.38 (*p* = 0.002) for those older than 75 years [219]. High rates of AKI development are associated with the use of penicillins (piperacillin tazobactam, cloxacillin, flucloxacillin) and vancomycin. A combination of piperacillin tazobactam with vancomycin was associated with a significantly higher incidence of AKI compared with piperacillin tazobactam plus meropenem combination (16.5% vs 3.6%; *p* = 0.009). Finally, piperacillin tazobactam with vancomycin was associated with a 6.8-fold increased risk of developing AKI (OR: 6.8, 95% confidence interval [CI] 1.5–30.9), and the higher plasma concentration of vancomycin was also a determinant of AKI risk [217]. A systematic review and meta-analysis of observational studies (12 studies included, 14,511 patients) revealed significantly higher odds of AKI development in patients treated with a combination of vancomycin plus piperacillin tazobactam compared with vancomycin plus meropenem combination (OR = 2.31; 95%CI 1.69–3.15) [237,238]. Similar results were demonstrated in another study (retrospective cohort study, period 20134—2019 years), where AKI incidence was reported to be 33.3% in patients receiving vancomycin with piperacillin tazobactam compared with 9.1% for those who received vancomycin with meropenem or doripenem [238]. Estimating AKI risks for meropenem, it is worth noting that the comparison of vancomycin plus meropenem versus vancomycin plus cefepime revealed a nearly 2-fold increase in AKI incidence for the first combination (38% versus 19.1%; *p* = 0.049) [239].

The highest AKI odds associated with piperacillin tazobactam were demonstrated in another study, with OR = 1.89 (95% CI: 1.73–2.06) [240]. Considering vancomycin monotherapy, overall incidence of AKI was 9.3 (95% CI 0.78–1.22) per 100 person-years, and the adjusted hazard ratio versus all other comparator antibiotics was 0.74 (95% CI: 0.45–1.21) [241]. Comparison of nephrotoxicity induced by glycopeptides revealed less AKI risks for teicoplanin compared with vancomycin (relative risk, RR = 0.66; 95% CI: 0.48–0.90; I2 = 10%) as it was established in the Cochrane systematic review [242], suggesting possible benefits of its inclusion in the combined AB schemes instead of vancomycin. This benefit was proved by the recent study, which revealed that piperacillin tazobactam with vancomycin combination compared with piperacillin tazobactam plus teicoplanin or vancomycin plus meropenem was associated with 3.96 times (95% CI, 1.48–10.63, *p* = 0.006) and 3.11 times (95% CI, 1.12–8.62; *p* = 0.028) increased risk of AKI, respectively [243].

The incidence rate of linezolid-induced AKI is considered to be lower than that of vancomycin, though there are results of a small study (63 patients with vancomycin and 38 with linezolid) indicating no difference in risk of AKI between these groups (*p* = 0.773). AKI occurred in 19 (30.2%) patients from the vancomycin group and in 14 (36.8%) patients from linezolid groups (*p* = 0.448) [223].

Considering imipenem, it is important to note the protective effect of cilastatin, and favorable outcomes for a combination of imipenem/cilastatin with relebactam compared with colistin adding. For the first antibiotics combination, AKI incidence was zero, for the second—31.3% [244].

Aminoglycosides are a common reason for nephrotoxicity. Amikacin-induced AKI in mechanically ventilated critically ill patients with sepsis was 26.7%, and among factors independently associated with an increased risk of amikacin-induced AKI were concurrent use of colistin (OR = 25.51, 95%CI: 6.99–93.05, *p* < 0.001), presence of septic shock (OR = 4.22, 95%CI: 1.76–10.11, *p* = 0.001), and Charlson Comorbidity Index (OR = 1.14, 95%CI: 1.02–1.28, *p* = 0.025) [226].

Analysis of the Food and Drug Administration Adverse Event Reporting System (FAERS) database from 2000 to 2021 year revealed antibacterial agents associated with AKI in older adults. The highest reporting odds ratios (ROR) were determined for the next ones: vancomycin (5.73 (95% CI: 5.30–6.21)), sulfamethoxazole (5.30 (95% CI: 4.80–5.85)), trimethoprim (5.25 (95% CI: 4.27–6.45), colistin (5.11 (95% CI: 3.17–8.22)), amoxicillin (2.75 (95% CI: 2.50–3.04)), ciprofloxacin (2.66 (95% CI: 2.45–2.89)), clarithromycin (2.75 (95% CI: 2.46–3.07)) [215]. AKI occurred in 68.5% of 412 enrolled patients with an incidence rate of 10.6 per 100 patients-days and a median time was 6 (3–13) days. Stages I–III of AKI were 38.3, 24.5, and 37.2%.

Estimation of patients with COVID-19 treated with antibiotics revealed a significantly higher incidence of AKI in those who received linezolid (*p* < 0.0001), vancomycin (*p* < 0.0001), carbapenem (*p* < 0.0001), cephalosporin (*p* < 0.0001), and piperacillin/tazobactam (*p* = 0.028). AKI was associated with prolonged hospitalization (OR = 3.37; 95% CI: 1.76–6.45) [245].

Antibiotic-induced AKI is also associated with increased mortality, especially in the elderly, as was demonstrated for patients who used intravenous colistin (hazard ratio, HR = 1.74, 95% CI: 1.06–2.86, *p* = 0.028). Colistin-induced AKI incidence rate was estimated as 10.6 per 100 patients-days, stage 1 was seen in 38.3%, stage 2 in 24.5%, and stage 3 in 37.2% [235].

Patients with severe malnutrition accompanied by dehydration are at increased risk of diminished glomerular filtration rate (GFR), renal blood flow decline, and impaired tubular excretion and reabsorption [43]. The existing evidence suggests that elimination of streptomycin may decrease and that of rifampicin increases in malnourished adults [43].

Hepatic impairment may directly or indirectly decrease protein binding, metabolism, and renal elimination of antibiotics [133]. Dose adjustment is needed for drugs that undergo hepatobiliary clearance, especially those that undergo phase I metabolism, have high protein binding, or are associated with high hepatotoxicity [133]. Liver cirrhosis has a significant impact on antibiotics disposition due to numerous pathological changes including liver cell necrosis, portosystemic shunt, reduction in the concentration of drug-binding proteins, atypical *Vd*, altered metabolism and elimination, altered PD, drug-drug interactions, and frequent association with renal failure) [246]. The percentage of antibiotics bound by albumin may be altered in cirrhotic patients [133]. Elimination of tigecycline which is lipophilic and highly protein bound (71–89%) demonstrated a reduction in patients with hepatic failure, accompanied by a 43% increase in elimination half-life in severe hepatic impairment [247]. In patients with severe liver disease, it is recommended to half the standard maintenance dose of tigecycline but no changes in its usual loading dose are needed [247].

Decompensated hepatic failure causes renal vasoconstriction and subsequent renal failure, leading to the reduction of renally eliminated antibiotics excretion and an increase of their serum concentrations [248]. Diminished renal elimination in liver cirrhosis was shown for ofloxacin, ampicillin, aminoglycosides, and vancomycin [246].

Aging and age-related diseases may affect the levels of circulating proteins modifying their renal excretion. The work by Lind L et al. (2019) revealed inverse relation of the change in eGFR to the change in most of the evaluated plasma proteins (74%), among which the most significant inverse relationships were reported for cystatin-B (CSTB), tumor necrosis factor receptor 1 (TNF-R1), CD40L receptor (CD40), tumor necrosis factor receptor 2 (TNF-R2), TNF-related apoptosis-inducing ligand receptor 2 (TRAIL-R2); study population included persons aged 70 at baseline, the study period was 10 years. A positive relationship was revealed between the change in eGFR and the change in hemoglobin (beta 0.10, SE 0.03, Pearson’s correlation coefficient 0.11, *p*-value = 7.9 × 10^−4^) [249].

Reduction of renal clearance mediated by different reasons leads to the increase in the half-life period of various drugs and indicates the need to decrease the daily dose of some antibiotics [213]. This is of paramount importance for antibiotics with narrow therapeutic index (NTI) including glycopeptides, aminoglycosides, and chloramphenicol succinate [213,250]. The risk of toxicity of NTI antibiotics is extremely increased in the elderly, especially in those with frailty [251], suggesting the actual need for therapeutic drug monitoring (TDM) [252].

Age-related changes in AB PK may be illustrated with data derived from linezolid TDM. Comparison of TDM samples from adult patients (<50 years) and from the elderly (>90 years) revealed a highly significant, progressive increment in the linezolid trough concentrations (5.8 ± 5.6 mg/L versus 16.6 ± 10.0 mg/L), an overall increment was 30% per decade of age. Increased trough concentrations contribute to the increased overdose and toxicity risks; they were found to exceed therapeutic levels in 30%, 50%, and 65% of patients aged <65 years, 65–80 years, and >80 years, respectively [253].

Investigation of the efficacy and safety of vancomycin in patients ≥ 80 years revealed a failure of treatment in 34.4%. The increased trough concentrations of vancomycin (VTC) were associated with increased 30-day mortality rates: for VTC at <10 μg/mL mortality rate was 2.8%, at 10 to 15 μg/mL—15.0%, at 15 to 20 μg/mL—15.3%, at ≥20 μg/mL—37.8%. The multivariate analysis determined blood urea nitrogen ≥ 11 g/dL and heart failure as independent factors associated with treatment failure (*p* = 0.004, 0.016, respectively). Nephrotoxicity was observed in 12.0% of patients treated with vancomycin. Independent factors associated with increased nephrotoxicity were VTC ≥ 15 μg/mL; treatment duration ≥ 15 d; and concomitant aminoglycosides administration (*p* = 0.024, 0.035, 0.029, respectively) [254]. Comparison of the VTC and AUC/MIC in the patients with the mean age (+standard deviation) 50.9 ± 12.4 versus 76.9 ± 8 years revealed their significant increase in the elderly. Rapid achievement of VTC ≥ 15 mg/L (within 4 days) was significantly more specific for the elderly compared with younger patients (54.1% vs. 36.5%, *p* = 0.004) as did 30-day mortality (40.9% vs. 12.5%, *p* < 0.001) [255]. In the work by Hatti M et al. (2018) a considerable variation of trough AB concentrations in older adults was demonstrated for cefotaxime, meropenem, and piperacillin-tazobactam, which was mainly related to the low eGFR. Increased trough concentrations of cefotaxime were significantly associated with older age, diabetes with end organ damage, moderate/severe kidney disease, and higher sepsis severity [256].

For beta-lactams renal function is an important factor affecting PK parameters and creatinine clearance was reported to be the most significant covariate altering beta-lactams PK in late elderly patients. Population PK methods revealed decreased levels of clearance for both piperacillin and tazobactam compared with younger population [257]. Doripenem in the elderly with nosocomial pneumonia was characterized by increased AUC and prolonged *T1/2*, reflecting a decrease in the renal clearance related to aging [258]. The same tendency was demonstrated for meropenem in the elderly: clearance was significantly lower than in younger patients due to the decline of renal function [259].

Aminoglycosides are also among antibiotics whose PK is dramatically affected by renal function decrease. Elimination of amikacin was delayed with increasing age, reflecting glomerular filtration rate decline [260].

Population pharmacokinetic modeling for levofloxacin in patients with a mean age of 81.2 years and impaired renal function demonstrated decreased mean clearance compared to healthy volunteers [261].

Another concern regarding renal function effects on PK is augmented renal clearance (ARC), defined as a creatinine clearance of more than 130 mL/min/1.73 m^2^ [262]. ARC is mediated by the changes in kidney function arising in a critically ill state. The meta-analysis of 70 studies revealed a pooled prevalence of ARC of 39% (95% CI: 34.9–43.3) and its risk factors in populations with apparently normal renal function including young age, male sex, and trauma [263]. In critically ill patients with cancer ARC development on the first day of intensive care unit (ICU) admission demonstrated a significant association with younger age (OR 1.028, 95% CI: 1.005–1.051) [264]. Despite the association with young age stated above a retrospective study of 2592 critically ill patients admitted to the ICU revealed that the median age of patients with ARC was 70 (55–79) years, with a prevalence of 33.4% [265], pointing out the importance of the ARC for the elderly population. This state can lead to decreased AB exposure and thus to treatment failure with beta-lactams, aminoglycosides, glycopeptides, and other, mainly hydrophilic agents [262]. Vancomycin use in ARC patients demonstrated achievement of the trough concentration in only 19.23% [266], and the same trend was revealed in the vancomycin population pharmacokinetic model, affirming that ARC was significantly associated with subtherapeutic serum concentrations [267]. Renal function is a significant predictor of proper meropenem exposure. In critically ill septic patients with ARC (median age 63 years, interquartile range, 55 to 68 years) poor PK/PD target attainment was demonstrated [268]. Linezolid clearance demonstrated a significant increase in ARC patients, resulting in sub-therapeutic concentrations after standard doses [269].

Age-related changes in the main PK parameters of antibacterial agents compared to the younger population are given in Table 7, Table 8, Table 9, Table 10, Table 11, Table 12 and Table 13.

## 4. AB Dosing Regimens in the Elderly

The main aim of AB therapy in elderly patients is to provide a proper balance between efficacy (PK/PD target attainment) and safety. Changed PK parameters may lead both to decreased or increased AB exposure contributing to negative treatment outcomes. Dose adjustment is a typical approach in the management of the elderly with infections. Decreased metabolizing capacity and declined renal clearance result in the need to decrease the standard adult dose of AB, while ARC specific for critically ill patients may dictate the necessity to use a higher dose.

Table 14, Table 15, Table 16, Table 17 and Table 18 include information about the proposed regimens of AB dosing depending on age, renal function, and hepatic function along with data on concentrations reported to cause toxic reactions.

## 5. Conclusions

Optimization of the management of elderly patients with infectious diseases is a complex process, and successful performance demands knowledge of the main age-related and pathology-related changes in the patient’s organism. Altered PK parameters may contribute to the decreased efficacy of the treatments with suboptimal antibiotic exposure or to the increased risks of toxic reactions ameliorating further response to drugs with overexposure. Individualization of the pharmacotherapy based on the unique characteristics of the elderly patients may ensure the attainment of an optimal PK/PD target and treatment success. The existing level of evidence on PK changes in the elderly clearly indicates a significant difference in most PK parameters compared to younger adults. The last decade is characterized by a tendency to increase the participation of the elderly in clinical trials. However, the number of such trials is still insufficient to cover all the classes of ABs and to provide full evidence-based background to choose proper dosing regimens in all the pathologic states specific to the elderly and senile patients. Global aging indicates an urgent need to extend inclusion of the elderly and senile patients with various comorbidity profiles and geriatric syndromes in clinical trials and PK studies.

## Figures and Tables

**Figure 1 biomedicines-11-01633-f001:**
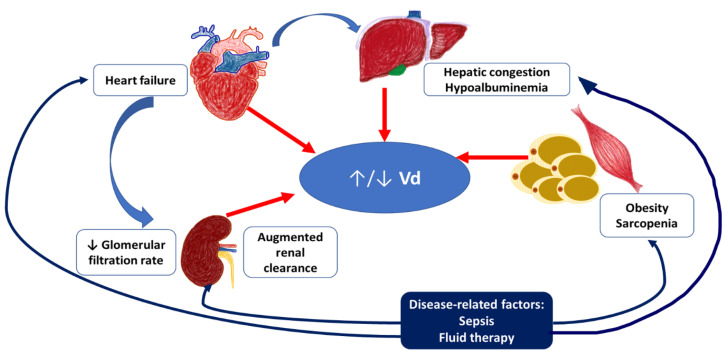
The list of the main factors affecting *Vd* and clearance in the elderly with severe infections and their interrelationships. Severe infections may worsen heart failure and hepatic congestion, contribute to sarcopenia progression and fat redistribution. Exacerbation of heart failure leads to the decreased renal function, while septic changes may result in the augmented renal clearance. In critically ill patients Vd of hydrophilic agents may be increased with hemodynamic insufficiency development, excessive permeability of capillaries, and infusion therapy, while Vd of lipophilic agents may be decreased due to the decrease of fat tissue mass. These changes are different from those observed in a healthy elderly patient (typically decreased Vd of hydrophilic agents and increased Vd of lipophilic agents).

**Table 1 biomedicines-11-01633-t001:** Senescence-related changes of GIT, drug-induced changes of GIT and their effects on PK.

Part of GIT	Age-Associated Changes	Drug-Induced Changes	PK Changes	References
Oral cavity	Xerostomia, dysgeusia and ageusia, oropharyngeal dysphagia	Xerostomia may be induced by cholinolytics, histamine H1 antagonists, α1 adrenergic antagonists, tricyclic antidepressants	Decreased absorption from the oral cavity	[45,46,47]
Esophagus	Esophageal dysphagia, odynophagia, increased risk of gastroesophageal reflux disease, Barrett’s columnar-lined esophagus, DIE	DIE may be induced by antibiotics (tetracycline, doxycycline, clindamycin), bisphosphonates (alendronate), calcium channel blockers (amlodipine), anti-coagulants (dabigatran, apixaban), Chemotherapeutic agents (sunitinib, doxorubicin, methotrexate, nivolumab, ipilimumab), ferrous sulfate, NSAIDs	Decreased absorption of weak acids and weak bases, high risk of drug interactions resulting in further PK changes	[45,48,49,50,51,52,53,54,55,56]
Stomach	Chronic atrophic gastritis, increased risks of hypochlorhydria and of hyperchlorhydria with peptic ulcer, decreased gastric motility.	Proton pump inhibitors (PPIs) contribute to the development of hypochlorhydria and may induce enterochromaffin-like cells hyperplasia, gastric polyp formation, and hypergastrinemia, PPI could increase the risk of community-acquired pneumonia, autoimmune diseases, cardiovascular diseases, onset of dementia and depression, fragility fractures, mainly hip fractures	Hypochlorhydria may result in the impaired drug dissolution and changed systemic exposure of poorly water-soluble drugs	[8,45,57,58,59,60,61,62,63]
Intestine	Malnutrition, chronic constipation, high risk of colorectal cancer, increased gut permeability, increased chronic and systemic mild inflammatory responses with risks for inflammatory bowel disease, dysbiosis (50% of microbiome in the elderly—Bacteroides, Alistipes, and Parabacteroides, versus 8–27% in a younger cohort), decreased small bowel surface area, increased rates of Clostridium difficile colitis, and diverticular disease	Drug-induced colitis may be caused by diuretics, dihydropyridines, glycosides, platelet aggregation inhibitors, NSAIDs, statins and fibrates, as well, as immune checkpoint inhibitors (ipilimumab and nivolumab), idelalisib, mycophenolate mofetil. PPIs are associated with the risk of developing Clostridium difficile infections	Decreased absorption mainly because of the decreased intestinal blood flow and decreased absorption area	[64,65,66,67,68,69]
Pancreas	Decreased pancreatic secretion (decreased lipase, chymotrypsin, amylase levels), pancreatic atrophy, lobulation, and fatty degeneration	The highest number of drug-induced pancreatitis cases were associated with the use of valproic acid, L-asparaginase, and 5-aminosalicylic acid	Fat malabsorption may alter absorption of lipophilic drugs	[70,71,72,73]
Liver	Liver volume decreases by 20–40% with aging, blood flow decrease by 35% compared with persons < 40 years old. Increased rates of oxidative stress and inflammatory response, high prevalence of liver fibrosis, NAFLD	Drug induced liver injury is commonly caused by antibacterials (amoxicillin-clavulanate, flucloxacillin, nitrofurantoin), statins (atorvastatin), immune checkpoints inhibitors (nivolumab, ipilimumab, infliximab)	Decreased first-pass metabolism with consequent increase of absorption and bioavailability of high-clearance drugs, decreased rates of formation of active drugs from prodrugs with consequent decrease of their plasma concentrations and possible failure of treatment	[74,75,76]

DIE—drug-induced esophagitis, PPI—proton pump inhibitors, NAFLD—nonalcoholic fatty liver disease, NSAIDs—non-steroidal anti-inflammatory drugs.

**Table 2 biomedicines-11-01633-t002:** Age-associated changes of transport proteins and data on their antibiotic-substrates.

Transport Protein	Age-Dependent Change of Expression	Transported Antibiotics	References
BCRP	Decreased protein expression in the human intestine and liver in the elderly. Gene expression is significantly decreased in animal models. No marked changes in human with aging	Fluoroquinolones (delafloxacin, ciprofloxacin, enrofloxacin, norfloxacin, ofloxacin), nitrofurantoin	[81,82,85,89,90,91]
P-gp	No marked changes in human with aging. Significant reduction of activity in the elderly with renal failure.	Erythromycin, tetracycline Azithromycin Levofloxacin, sparfloxacin Dicloxacillin	[81,92,93,94,95,96,97]
MRP2	No data available in human	Ampicillin, azithromycin, ceftriaxone, cefodizime, ceftriaxone	[98]
OATP1A2	No data available in human	Most of fluoroquinolones (ciprofloxacin, enoxacin, gatifloxacin, levofloxacin, lomefloxacin, norfloxacin), erythromycin, tebipenem	[93,99,100,101]
OATP1B1	No marked changes in human aging. A weak correlation was noted between OATP1B1 abundance and age of human donors. Significant reduction of activity in the elderly with renal failure	Benzylpenicillin, rifampicin, rifampin, rifampicin, cefazolin, cefditoren, cefoperazone, nafcillin	[81,85,86,93,102,103]
OATP1B3	No marked changes in human aging	Rifampicin, rifampin, cefadroxil, cefazolin, cefditoren, cefmetazole, cefoperazone, cephalexin, nafcillin, erythromycin	[86,93,100,103]
OATP2B1	No marked changes in human aging. Gene expression is significantly decreased in animal models	Benzylpenicillin, tebipenem pivoxil	[86,88,104,105]
PEPT1	In diabetes mellitus—downregulation of PepT1. In obesity—leptin-dependent activation of PepT1 activity and expression	Penicillins (penicillin G, cyclacillin), cephalosporines (cefadroxil, ceftibuten, cefixime, cephradine, cephalexin, cefroxadine, loracarbef)	[83,103]
PEPT2	Age-dependent changes were observed for different locations (heart, brain, nervous tissue, kidney) with increase of expression with aging	Colistin	[84,106,107]

**Table 3 biomedicines-11-01633-t003:** Food effects on PK parameters of antibiotics.

Drug (Oral Administration)	Food Effect on PK Parameters	References
Ampicillin	Plasma concentration is decreased in fed state. Should not be taken with food to allow optimal absorption	[110]
Amoxicillin	*Cmax* decreased, *Tmax* prolonged under fed condition, but with no change of the AUC, thus use both under fasted and fed state is effective, since it is time-dependent AB	[111]
Amoxicillin-clavulanate	Decreased bioavailability of clavulanic acid after meal (extended-release tablets), so administration before meal is preferrable	[112]
Flucloxacillin	Reduced *AUC*, *Cmax*, and prolonged *Tmax* of both free and total concentrations compared with the fasting state. Achievement of free concentration targets associated with efficacy was in most circumstances equivalent, suggesting no negative association with the fed state	[113]
Cefaclor	*Cmax* decreased, *Tmax* delayed, but no *AUC* changes were reported for cefaclor granule and cefaclor suspension under fed state supposing effective use regardless of meal	[114]
Cefuroxime axetil	Positive food effect on absorption with *AUC* in fed state greater than in the fasted state, suggesting postprandial administration be more effective	[109]
Cefpodoxime proxetil	Achievement of proper *Cmax* and MIC values was reported in non-fasting patients. In the elderly patients, the absorption is approximately 30% lower compared to younger patients	[115,116,117]
Azithromycin	Capsules have delayed disintegration under fed state, resulting in the extended gastric residence and gastric degradation of azithromycin, thus capsules should be taken only in the fasted state. Tablets can be used regardless of meal	[118]
Clindamycin	The extent to systemic exposure was affected by the delay in absorption in the fed state, suggesting optimal dosing is in the fasted state	[119]
Linezolid	A slight decrease in *Cmax*, and delay in *Tmax* were observed in fed state, with no effect on *AUC*, suggesting effective use regardless of meal	[120]
Ofloxacin	*Cmax* and *AUC* were greater in the fasted state, significant decrease of absorption was observed with aluminum co-administration	[121]
Ciprofloxacin	*Cmax* and *AUC* were greater in the fasted state, significant decrease of absorption was observed with aluminum co-administration. Meal should be held for 1 h before and 2 h after fluoroquinolone administration	[121,122]
Levofloxacin	Slight delay of absorption with no alteration of the overall bioavailability after high fat meal. Food or drinks enriched with calcium may decrease *Cmax* and delay *Tmax*	[123,124]
Moxifloxacin	Considerable decrease of plasma concentrations in the fed state in comparison with the fasted state, so preprandial use is recommended	[125]
Doxycycline	Decrease of the *Cmax* and *AUC* in the fed state compared with the fasted state	[126]

**Table 4 biomedicines-11-01633-t004:** Age-associated changes of absorption for non-oral routes of drugs administration in the elderly.

Route of Administration	Age-Associated Change	Absorption Change	References
Intramuscular	Sarcopenia (the loss of muscle mass and function), fibrosis, infiltration of fat into skeletal muscle, increased inflammatory response	Some increase for depot preparations	[127]
	Hypoperfusion of skeletal muscles	May decrease	[76,128]
Percutaneous	Decreased hydration and changed lipid structure result in an increased barrier function of the stratum corneum	Some decrease for hydrophilic drugs	[129]
Inhalation	Lung function decline: increase in alveolar size and alveolar-capillary surface area, reduction of the elastic recoil of the lungs, increase in end-expiratory lung volume, increase of the functional residual capacity, reduction of the expiratory airflow, decline in forced expiratory volume by approximately 30 mL/year and forced vital capacity by approximately 20 mL/year, decrease of the blood flow rates	Variable effect on absorption, leading to increase, decrease, or no changes of AUC and Cmax compared to younger patients	[130,131]

**Table 5 biomedicines-11-01633-t005:** Substrates, inhibitors, and inducers of CYP450 isoenzymes among antibacterial agents and CYP450 aging changes.

CYP450 Isoenzyme	Antibiotic-Substrate	Antibiotic-Inducer or Antibiotic-Inhibitor	Age-Related Changes	References
CYP1A1	Linezolid	Inhibitor-Norfloxacin	Certain change is unknown. CYP1A1 polymorphism is supposed to be related to the development of multiple age-associated diseases (cancers, chronic obstructive pulmonary disease, coronary artery diseases)	[165,166,167]
CYP1A2	Grepafloxacin, Lomefloxacin	Inhibitors—quinolones and fluoroquinolones Inducers—Rifampicin, Nafcillin	Certain change is unknown. CYP1A2 polymorphism is supposed to be related to the development of multiple age-associated diseases (cancers, hypertension, chronic obstructive pulmonary disease, coronary artery diseases)	[167,168,169,170]
CYP2A6	Metronidazole	Inhibitors—Isoniazid, Ethambutol Inducers—Rifampicin	Weak positive association of the age and CYP2A6 protein levels and enzyme activity (nicotine and coumarin metabolism studies)	[171,172,173,174]
CYP1B1	Linezolid	NA	Age-related changes are supposed. High frequency expression along with polymorphism is specific for a variety of cancers, obesity, glucose intolerance. CYP1B1 is involved in hypertension development and progression	[165,166,167,168,169,170,171,172,173,174,175]
CYP2B6	NA	Inhibitors—Rifamycin Inducers—Rifampicin, Rifabutin, Rifamycin, Rifapentine	Age modified the effect of CYP2B6 genotype on loss to care in older HIV positive Africans: older slow metabolizers were at over four-fold higher risk when compared to older intermediate metabolizers (OR: 4.06 95% CI: 1.38, 11.89)	[176,177,178]
CYP2C8	Linezolid, Trimethoprim	Inhibitors—Trimethoprim, Metronidazole, Isoniazid, Rifampicin, Rifamycin, Amoxicillin Inducers—Rifampicin, Rifabutin, Rifamycin, Rifapentine, Rifaximin	Some decrease is supposed. CYP2C8 provides anti-inflammatory and anti-oxidative effects in the vessels, its induction leads to the suppression of TNF-α induced inflammatory cytokines	[165,178,179,180,181,182]
CYP2C9	Sulfamethoxazole, Trimethoprim	Inhibitors—Metronidazole, Sulfamethoxazole, Isoniazid, Sulfadiazine, Sulfisoxazole, Sulfamethizole, Rifamycin, Oritavancin Inducers—Rifampicin, Rifapentine, Rifabutin, Rifamycin	Systemic celecoxib exposure suggests that for the elderly extensive metabolizers enzyme activity may exceed that of younger ones. For intermediate and poor elderly metabolizers activity is reduced compared to the young ones. Systemic warfarin exposure was higher in all types of elderly metabolizers compared to young ones	[85,183,184]
CYP2C19	NA	Inhibitors—Chloramphenicol, Oritavancin, Isoniazid, Sulfanilamide, Rifamycin, Ethambutol Inducers—Rifampicin, Rifamycin, Rifapentine, Rifabutin, Rifaximin	A decline in CYP2C19 metabolic activity was associated with sarcopenia and fatty liver disease in the elderly	[162,185,186]
CYP2D6	Linezolid, Fusidic acid	Inhibitors—Isoniazid, Fusidic acid, Rifamycin, Oritavancin, Ethambutol	Decrease is supposed due to 20% reduction for CYP2D6 substrates. Less activity of CYP2D6 was in poor metabolizers >65 years compared with those <40 years (*p* < 0.001)	[23,165,187,188,189]
CYP2E1	Isoniazid	Inhibitors—Isoniazid Inducers—Delafloxacin, Isoniazid, Rifampicin	Significant reduction of the protein levels was observed in liver pathology	[163,190,191,192]
CYP3A4	Erythromycin, Linezolid, Clindamycin, Telithromycin, Clarithromycin, Azithromycin, Rifabutin, Rifapentine, Rifaximin, Grepafloxacin, Roxithromycin, Cethromycin, Clindamycin, Tetracycline, Trimethoprim, Cephalexin, Sulfadiazine, Fusidic acid, Eravacycline, Flucloxacillin	Inhibitors—Macrolides, Isoniazid, Dalfopristin, Quinupristin, Chloramphenicol, Metronidazole, Fusidic acid, Clindamycin, Ciprofloxacin, Norfloxacin, Tetracycline, Doxycycline, Sulfamethoxazole, Sulfanilamide, Rifamycin, Oritavancin Inducers—Rifabutin, Rifampicin, Rifapentine, Rifaximin, Rifamycin, Nafcillin, Oritavancin, Flucloxacillin, Dicloxacillin, Cefradine, Delafloxacin	Decrease is supposed due to 30 to 50% clearance reduction for the CYP3A4 substrates Decline of CYP3A4 activity was associated with sarcopenia in the elderly	[24,162,165,193,194,195,196,197]
CYP3A5	Linezolid, Clindamycin, Clarithromycin, Telithromycin, Cethromycin, Erythromycin, Metronidazole, Clindamycin	Inhibitors—Ciprofloxacin, Erythromycin, Clarithromycin, Telithromycin, Chloramphenicol, Inducers—Rifampicin	Excessive systemic substrate exposure suggests decline of activity in the elderly	[85,165,198,199,200,201]
CYP3A7	Clarithromycin, Erythromycin, Telithromycin, Metronidazole	Inhibitors—Erythromycin, Ciprofloxacin, Norfloxacin, Chloramphenicol Inducers—Rifampicin	Primarily expressed in the fetus and newborn, with relative decline with aging	[202,203,204,205,206]

**Table 6 biomedicines-11-01633-t006:** Antibacterial agents associated with AKI development and mechanisms of their nephrotoxic effects.

Antibacterial Agents	Mechanism of AKI	References
Amoxicillin, Flucloxacillin, Piperacillin−tazobactam, Cloxacillin, Nafcillin	AIN with a proposed role of allergic inflammation	[215,216,217,218,219,220]
Cefazolin, Ceftriaxone, Cefepime	AIN with a proposed role of allergic inflammation	[221,222]
Vancomycin	Dose-dependent induction of oxidative stress, complement activation, and mitochondrial damage resulting in the acute tubular injury/necrosis or acute tubulointerstitial nephritis. New mechanism—drug-induced obstructive tubular cast formation. Acute tubulointerstitial nephritis with significant eosinophil infiltration, suggesting allergic mechanism	[216,221,223,224,225]
Linezolid	AIN	[223]
Gentamicin, amikacin	Apical transport results in the accumulation of aminoglycosides within tubular cells leading to the cell injury and death (proximal tubulopathy) due to lysosomal accumulation, inhibition of lysosomal enzymes and formation of myelin bodies. direct proximal and distal tubule cytotoxicity	[221,226]
Clarithromycin	Cell-mediated hypersensitivity reaction resulting in acute kidney injury and nephrotic syndrome. Drug interaction: macrolides are CYP3A4 inhibitors, their concomitant use with calcium blockers may result in excessive hypotension leading to the ischemic acute kidney injury	[215,227,228]
Ciprofloxacin Levofloxacin	Crystal-induced acute kidney injury, damage of the collecting duct. Urine pH more than 6.0 mediates crystal precipitation within tubular lumens	[224,229,230,231]
Sulfamethoxazole and trimethoprim	Intrinsic renal impairment, sulfamethoxazole urine crystal formation	[215,232,233,234]
Colistin	Accumulation of colistin in the proximal tubule cells, direct targeting the mitochondria	[215,235,236]

**Table 7 biomedicines-11-01633-t007:** Comparative data on PK parameters of β-lactam antibiotics in the elderly and adults.

Drug		Volume of Distribution, Vd		Plasma Protein Binding Rate, PPB Rate		Clearance, CL		Half-Life Period, T1/2		References
		Elderly Patients	Adults	Elderly Patients	Adults	Elderly Patients	Adults	Elderly Patients	Adults	
Penicillins										
Amoxicillin		21 ± 9 L (mean age 82 ± 6 years)	20 L	NA	20%	109 ± 72 mL/min (mean age 82 ± 6 years)	230–280 mL/min	2.4 ± 0.6 h (i.v.), 2.0 ± 0.41 h (capsule), 1.9 ± 0.55 h (solutab) (mean age 82 ± 6 years)	1 h	[270,271]
Ampicillin/Sulbactam	Ampicillin	26.33 ± 8.75 L (mean age 73.9 ± 5.1 years) 19.3 ± 0.2 L (mean age 85.7 ± 7.9 years)	31.4 ± 13.12 L (mean age 30 ± 6.5 years) 31.29 + 8.72 L (mean age 51 ± 7.3 years)	NA	18–28%	198.02 ± 55.60 mL/min (mean age 73.9 ± 5.1 years) 6.5 ± 4.0 L/h (mean age 85.7 ± 7.9 years)	289.15 ± 50.52 mL/min (mean age 30 ± 6.5 years) 281.29 ± 33.64 mL/min (mean age 51 ± 7.3 years)	1.35 ± 0.29 h (mean age 73.9 ± 5.1 years) 2.7 ± 1.6 h, (mean age 85.7 ± 7.9 years)	0.86 ± 0.15 h (mean age 30 ± 6.5 years) 1.09 ± 0.18 h (mean age 51 ± 7.3 years)	[272,273]
	Sulbactam	23.54 ± 7.71 L (mean age 73.9 ± 5.1 years) 18.6 ± 6.8 L (mean age 85.7 ± 7.9 years)	24.98 ± 4.66 L; (mean age 30 ± 6.5 years) 29.76 + 10.01 L (mean age 51 ± 7.3 years)		38%	162.69 ± 46.21 mL/min (mean age 73.9 ± 5.1 years) 5.6 ± 3.3 L/h (mean age 85.7 ± 7.9 years)	254.96 ± 53.04 mL/min (mean age 30 ± 6.5 years) 236.16 ± 26.98 mL/min (mean age 51 ± 7.3 years)	1.58 ± 0.29 h (mean age 73.9 ± 5.1 years) 3.3 ± 3.3 h (mean age 85.7 ± 7.9 years)	0.93 ± 0.15 h (mean age 30 ± 6.5 years) 1.19 + 0.17 h (mean age 51 ± 7.3 years)	
Cephalosporins										
Ceftaroline		Vss 17.9 ± 3.0 L (mean age 72.2 years)	Vss 15.8 ± 2.7 L (age range 18 to 45 years)	NA	20%	95.7 ± 13.4 mL/min (mean age 72.2 years)	127.3 ± 15.0 L mL/min (age: 18 to 45 years)	3.1 ± 0.4 h (mean age 72.2 years)	2.2 h (age: 18 to 45 years)	[274]
Cefepime		Vss 0.23 ± 0.03 L/kg (mean age 67 ± 2 years, (men)) Vss 0.24 ± 0.03 L/kg (mean age 69 ± 5 years, (women))	Vss 0.21 ± 0.02 L/kg (mean age 30 ± 6 years, men) Vss 0.21 ± 0.02 L/kg (mean age 33 ± 5 years, women)	NA	20%	1.11 ± 0.12 mL/min/kg (mean age 67 ± 2 years, men) 1.22 ± 0.19 mL/min/kg (mean age 69 ± 5 years, women)	1.54 ± 0.22 mL/min/kg (mean age 30 ± 6 years, men) 1.56 ± 0.22 mL/min/kg (mean age 33 ± 5 years, women)	3.05 ± 0.50 h (mean age 67 ± 2 years, men) 2.92 ± 0.38 h (mean age 69 ± 5 years, women)	2.26 ± 0.51 h (mean age 30 ± 6 years, men) 2.15 ± 0.33 h (mean age 33 ± 5 years, women)	[275,276]
Ceftriaxone		0.144 ± 0.018 L/ kg (mean age 69.6 ± 5.1 years)	8.5 ± 1.3 L (age range 19 to 40 years)	NA	83–96%	1.17 ± 0.29 L/h (mean age 69.6 ± 5.1 years)	0.68 ± 0.11 L/h (age 19 to 40 years)	6.9 ± 1.7 h (mean age 69.6 ± 5.1 years)	8.1 ± 0.3 h age 19 to 40 years:	[277,278,279]
Carbapenems										
Doripenem		median value 28.4 (IQR: 15.7–37.0) L (age >60 years)	16.8 L	NA	8.1%	median value 19.2 (IQR: 12.8–23.9) L/h (age > 60 years)	16.0 L/h	1.89 h (age >60 years)	1 h	[258,280,281]
Imipenem cilastatin	Imipenem	0.33 ± 0.09 L/kg (age 68 to 83 years)	Vc 0.16 ± 0.05 L/kg (age 19 to 34 years)	NA	20%	159.20 ± 48.38 mL/min/kg (age 68 to 83 years)	12.1 ± 0.06 L/h 1.73 m^2^ (age 19 to 34 years)	1.6 ± 0.72 h (age 68 to 83 years)	0.93 ± 0.09 h (age 19 to 34 years)	[282,283,284]
	Cilastatin	0.26 ± 0.07 L/kg (age 68 to 83 years)	Vc 0.14 ± 0.03 L/kg (age 19 to 34 years)			138.96 ± 81.6 mL/min/kg (age 68 to 83 years)	12.4 ± 1.1 L/h 1.73 m^2^ (age 19 to 34 years)	2.1 ± 2.14 h (age 68 to 83 years)	0.84 ± 0.11 h (age 19 to 34 years)	
Meropenem		Vc 17.2 ± 14 L Vp 10.6 ± 13 L (median age 75 (65–94) years) 13.2 ± 1.4 L/1.73 m^2^ (mean age 73 ± 4.6 years)	11.7 ± 1.2 L/1.73 m^2^ (mean age 28 ± 5.2 years)	NA	2%	5.27 L/h (median age 75 (65–94) years) 139 ± 20.0 mL/min 1.73 m^2^ (mean age 73 ± 4.6 years)	15.2 L/h	1.27 h (age 65 to 80 years) 1.27 h (mean age 73 ± 4.6 years)	0.81 h (mean age 28 ± 5.2 years)	[259,279,285,286]
Biapenem		4.19 ± 1.58 L (mean age 78.5 ± 5.3 years) Vss (dose 300 mg) 15.2 ± 4.1 L Vss (dose 600 mg) 15.1 ± 2.7 L (mean age 71.6 ± 2.7 years) Vss (dose 300 mg) 13.7 ± 2.7 L Vss (dose 600 mg) 13.4 ± 3.1 L (mean age 77.8 ± 1.9 years)	Vss 16.4 ± 2.64 L (dose 1250 mg) 15.3 ± 4.69 L (dose 1000 mg) 22.4 ± 8.55 L (dose 250 mg) (mean age 37.9 years)	NA	7%	6.22 ± 1.87 L/h (mean age 78.5 ± 5.3 years) 8.8 ± 1.1 L/h (dose 300 mg), 8.9 ± 1.9 L/h (dose 600 mg), (mean age 71.6 ± 2.7 years) 6.8 ± 0.9 L/h (dose 300 mg), 6.7 ± 1.2 L/h (dose 600 mg) (mean age 77.8 ± 1.9 years)	8.73 ± 1.99 L/h (dose 1000 mg), 14.2 ± 1.22 L/h (dose 250 mg), (average age 37.9 years)	1.82 ± 1.14 h (dose 300 mg), 1.45 ± 0.36 h (dose 600 mg), (mean age 71.6 ± 2.7 years) 1.75 ± 0.23 h (dose 300 mg), 1.59 ± 0.18 h (dose 600 mg), (mean age 77.8 ± 1.9 years)	1.03 ± 0.03 h (dose 750 mg), 1.31 ± 0.31 h (dose 1250 mg), (average age 37.9 years)	[287,288,289]

Vc—volume of the central compartment, Vss—volume in steady state, NA—not available.

**Table 8 biomedicines-11-01633-t008:** Comparative data on PK parameters of glycopeptides, lipopeptides, and lipoglycopeptides in the elderly and adults.

Drug	Volume of Distribution, Vd		Plasma Protein Binding Rate, PPB Rate		Clearance, CL		Half-Life Period, T1/2		References
	Elderly Patients	Adults	Elderly Patients	Adults	Elderly Patients	Adults	Elderly Patients	Adults	
Vancomycin	154 L (mean age 78.3 ± 6.96 years) 74.2 ± 32.3 L (age ≥ 60 years)	54.20 L (median age 37 (26–49.3) years)	NA	50%	2.45 L/h (mean age 78.3 ± 6.96 years) 0.71 ± 0.41 mL/min/kg (age ≥ 60 years)	7.29 L/h median age 37 (26–49.3) years:	17.8 ± 11.8 h (age ≥ 60 years)	4–6 h	[290,291,292,293,294]
Teicoplanin	Vc 78.1 (18.2) L (mean age 77.1 ± 11.4 years, men) 80.1 ± 7.0 years, women)	Vss 1.21 ± 0.56 L/kg (age range 19 to 31 years)	NA	90–95%	0.51 ± 3.9 L/h (mean age 77.1 ± 11.4 years, men 80.1 ± 7.0 years, women)	0.21 ± 0.018 mL/min/kg (age range 19 to 31 years)	106.1 h (mean age 77.1 ± 11.4 years, men 80.1 ± 7.0 years, women)	157 ± 92.8 h (age range 19 to 31)	[295,296,297]
Daptomycin	Vss 0.15 L/kg (age >75 years)	Vss 0.14 L/kg	NA	87–92%	9.86 mL/h/kg (age >75 years)	15.09 mL/h/kg	11.85 h (age >75 years)	6.79 h	[298]
Telavancin	Vss 156 ± 12 mL/kg (mean age 70.7 ± 5.6 years)	157 ± 19 mL/kg	NA	93%	12.2 ± 1.4 mL/min/kg (mean age 70.7 ± 5.6 years)	12 ± 2 mL/h/kg	9.3 ± 1.3 h (mean age 70.7 ± 5.6 years)	9.6 ± 2.9 h	[299,300]

Vc—volume of the central compartment, Vss—volume in steady state, NA—not available.

**Table 9 biomedicines-11-01633-t009:** Comparative data on PK parameters of oxazolidinones in the elderly and adults.

Drug	Volume of Distribution, Vd		Plasma Protein Binding Rate, PPB Rate		Clearance, CL		Half-Life Period, T1/2		References
	Elderly Patients	Adults	Elderly Patients	Adults	Elderly Patients	Adults	Elderly Patients	Adults	
Linezolid	0.61 ± 0.08 L/kg, (mean age 70.1 ± 3.4 years, men) 0.54 ± 0.13 L/kg, (mean age 69.9 ± 3.4 years, women)	0.77 ± 0.25 L/kg, (mean age 29.6 ± 7.1 years, men) 0.54 ± 0.17 L/kg, (mean age 29.5 ± 6.0 years, women)	NA	31%	CLPO 1.63 ± 0.44 mL/min/kg CLR 0.31 ± 0.06 mL/min/kg CLNR 1.31 ± 0.42 mL/min/kg, (mean age 70.1 ± 3.4 years, men) CLPO 1.30 ± 0.42 mL/min/kg CLR 0.36 ± 0.10 mL/min/kg CLNR 0.94 ± 0.47 mL/min/kg, (mean age 69.9 ± 3.4 years, women)	CLPO 1.67 ± 0.27 mL/min/kg CLR 0.44 ± 0.07 mL/min/kg CLNR 1.23 ± 0.25 mL/min/kg, (mean age 29.6 ± 7.1 years, men) CLPO 1.34 ± 0.33 mL/min/kg CLR 0.43 ± 0.09 mL/min/kg CLNR 0.91 ± 0.26 mL/min/kg, (mean age 29.5 ± 6.0 years, women)	4.6 ± 1.3 h, (mean age 70.1 ± 3.4 years, men) 5.3 ± 2.2 h (mean age 69.9 ± 3.4 years, women)	5.3 ± 1.7 h, (mean age 29.6 ± 7.1 years, men) 4.8 ± 1.5 h, (mean age 29.5 ± 6.0 years, women)	[301,302]
Tedizolid	mean age 71.9 ± 5.08 years: 91.6 ± 28.2 L	age 18 to 48 years: 95.7 ± 23.5 L	NA	70–90%	mean age 71.9 ± 5.08 years: 5.2 ± 1.6 L/h	age 18 to 48 years: 6.08 ± 1.08 L/h	mean age 71.9 ± 5.08 years: 12.3 ± 1.3 h	age 18 to 48 years: 11 h	[303,304]

CLPO—oral clearance, CLR—renal clearance, CLNR—non-renal clearance, NA—not available.

**Table 10 biomedicines-11-01633-t010:** Comparative data on PK parameters of tigecycline in the elderly and adults.

Drug	Volume of Distribution, Vd		Plasma Protein Binding Rate, PPB Rate		Clearance, CL		Half-Life Period, T1/2		References
	Elderly Patients	Adults	Elderly Patients	Adults	Elderly Patients	Adults	Elderly Patients	Adults	
Tigecycline	mean Vss 367 ± 96 L (mean age 65–75 years, women) mean Vss 499 ± 78 L (mean age 65–75 years, men) mean Vss 377 ± 123 L (mean age > 75 years, women) 401 ± 58 L (mean age > 75 years, men)	Vss 355 ± 95 L (mean age < 50 years, women) 554 ± 158 L (mean age < 50 years, men)	NA	71–89%	20.4 ± 4.7 L/h (mean age 65–75 years, women) 23.8 ± 4.3 L/h (mean age 65–75 years, women) 19.6 ± 3.6 L/h (mean age > 75 years, women) 18.7 ± 3.0 L/h (mean age 65–75 years, men)	<50 years: 20.6 ± 4.8 L/h (women) 28.5 ± 11.8 L/h (men)	16.5 ± 4.1 h (mean age 65–75 years, women) 19.5 ± 3.1 h (mean age 65–75 years, men) 21.2 ± 12.5 h (mean age > 75 years, women) 19.0 ± 5.0 h (mean age > 75 years (men)	17.1 ± 8.4 h (mean age < 50 years, women) 22.3 ± 15.3 h (mean age < 50 years, men)	[305,306]

Vss—volume in steady state.

**Table 11 biomedicines-11-01633-t011:** Comparative data on PK parameters of fluoroquinolones in the elderly and adults.

Drug	Volume of Distribution, Vd		Plasma Protein Binding Rate, PPB Rate		Clearance, CL		Half-Life Period, T1/2		References
	Elderly Patients	Adults	Elderly Patients	Adults	Elderly Patients	Adults	Elderly Patients	Adults	
Levofloxacin	Vc 52.95 ± 21.57 L (mean age 81.2 ± 5.08 years)	Vc 106 ± 12 L	NA	24–38%	2.53 ± 1.46 L/h (mean age 81.2 ± 5.08 years)	186 ± 5 mL/min	1.47 ± 0.65 h (mean age 81.2 ± 5.08 years)	6.91 ± 0.83 h	[261,307,308]
Moxifloxacin	2.24 L/kg (age range 69 to 81 years, men) 2.12 L/kg (age range 68 to 80 years, women)	2.60 L/kg (age range 22 to 44 years)	NA	40–50%	10.38 L/h (age range 69 to 81 years, men) 8.05 L/h (age range 68 to 80 years, women)	10.61 L/h (age range 22 to 44 years)	12.42 h (age range 69 to 81 years, men) 11.47 h (age range 68 to 80 years, women)	13.16 h (age range 22 to 44 years)	[309]
Ciprofloxacin	mean Vc 49.8 L mean Vp 63.3 L (mean age 70 ± 9 years)	2.00–3.04 L/kg		20–40%	mean CL 17.8 L/h (mean age 70 ± 9 years)	9.62 mL/min/kg	mean half-life 6.7 ± 4.1 h (mean age 70 ± 9 years)	4 h	[310,311]

Vc—volume of the central compartment, Vp—volume of the peripheral compartment, Vss—volume in steady state, NA—not available.

**Table 12 biomedicines-11-01633-t012:** Comparative data on PK parameters of aminoglycosides in the elderly and adults.

Drug	Volume of Distribution, Vd		Plasma Protein Binding Rate, PPB Rate		Clearance, CL		Half-Life Period, T1/2		References
	Elderly Patients	Adults	Elderly Patients	Adults	Elderly Patients	Adults	Elderly Patients	Adults	
Amikacin	18.0 ± 3.4 L (mean age 80.6 ± 7.3 years) 0.47 ± 0.14 L/kg (mean age 73.6 ± 9.1 years)	0.27 ± 0.06 L/kg	NA	≤10%	2.25 ± 0.78 L/h (mean age 80.6 ± 7.3 years) [309] 64.7 ± 42.7 mL/min (mean age 73.6 ± 9.1 years)	1.32 ± 0.55 mL/min/kg	5.8 ± 2.5 h (mean age 73.6 ± 9.1 years)	2.3 ± 0.44 h	[312,313,314,315]
Gentamicin	14.8 ± 1.4 L (mean age 80.4 ± 6.4 years, frail patients) 15.2 ± 2.2 L (mean age 80.4 ± 6.4 years, non-frail) 0.37 L/kg (age 70 to 96 years)	0.35 L/kg	NA	<20%	46.6 ± 10.7 mL/min (mean age 80.4 ± 6.4 years frail patients) 58.2 ± 12.4 mL/min (mean age 80.4 ± 6.4 years, non-frail) 1.0 mL/min/kg (age 70 to 96 years)	1.67 mL/min/kg	4.1 h (age 70 to 96 years)	2.5 h	[316,317]

NA—not available.

**Table 13 biomedicines-11-01633-t013:** Comparative data on PK parameters of polymyxin B in the elderly and adults.

Drug	Volume of Distribution, Vd		Plasma Protein Binding Rate, PPB Rate		Clearance, CL		Half-Life Period, T1/2		References
	Elderly Patients	Adults	Elderly Patients	Adults	Elderly Patients	Adults	Elderly Patients	Adults	
Polymyxin B	Vc 8.17 ± 0.67 L Vp 21.21 ± 8.28 L (age > 65 years) 0.490 ± 0.142 L/kg (age 63 to 73 years)	Vc 0.0929 L/kg Vp 0.330 L/kg	NA	92–99%	1.98 ± 0.67 L/h (age > 65 years) 0.028 ± 0.007 L/kg/h (age 63 to 73 years)	2.5 L/h	12.5 ± 3.11 h (age 63 to 73 years)	9–11.5 h	[318,319,320,321]

Vc—volume of the central compartment, Vp—volume of the peripheral compartment, NA—not available.

**Table 14 biomedicines-11-01633-t014:** Beta-lactam AB dosing depending on age, renal function, and hepatic impairment.

Drug	Regimen for Patients with Different Renal Function	PK/PD Target in the Elderly	Regimen for Patients with Hepatic Impairment	Safety	References
Penicillin group					
Ampicillin/Sulbactam	mean age > 65 years: 2 g of ampicillin/ 1 g of sulbactam every 8 h (normal renal function)	75–100 % *T* > MIC (MIC_90_ = 1 mg/L)	NA	Transient low-level elevations of ALT or AST in serum indicating transient liver damage	[322,323]
	mean age > 75 years: 1 g of ampicillin/ 0.5 g of sulbactam every 6 h (10 ≤ CLCR < 50 mL/min)	40% *T* > MIC (MIC = 8 μg/mL)			
Piperacillin/Tazobactam	mean age 85 (82–87) years: 4.5 g every 24 h (CLCR 0–19 mL/min/1.73 m^2^)	*fCss*/MIC ≥ 1 MIC ≤ 8 mg/L	4.5 g every 4–6 h (loading dose) 4.5 g every 6 h (maintenance dose)	Plasma concentration ≥ 157.2 μg/mL—risk of neurotoxicity	[324,325,326]
	mean age 85 (82–87) years: 9 g every 24 h (CLCR 20–39 mL/min/1.73 m^2^)				
	mean age 85 (82–87) years: 11.25 g every 24 h (CLCR 40–59 mL/min/1.73 m^2^)				
	mean age 85 (82–87) years: 13.5 g every 24 h (CLCR 60–79 mL/min/1.73 m^2^)				
Cephalosporins					
Cefepime	frail patients: 1 g every 12 h (CLCR = 30 mL/min)	*fT* > 50% MIC (susceptible strains)	1–2 g every 8–12 h (loading dose) 1–2 g every 8–12 h (maintenance dose)	Plasma concentration ≥ 38.1 mg/L—risk of neurotoxicity	[325,327,328]
	frail patients: 1 g every 8 h (CLCR 30–60 mL/min)				
	frail patients: 2 g every 8 h (normal renal function)	*fT* > 80% MIC (susceptible strains)			
Ceftriaxone	mean age > 65 years: 1 g every 48 h (eGFRcys 10 mL/min/1.73 m^2^) [326]	unbound fraction of ceftriaxone >MIC (MIC = 0.5–1 mg/L) [326]	1–2 g every 12 h (loading dose) 1–2 g every 12 h (maintenance dose) [322]	Plasma concentration ≥ 22 mg/L—risk of neurotoxicity and ceftriaxone-induced encephalopathy [327]	[329,330]
	mean age > 65 years: 2 g every 48 h (eGFR_CR-cys_ 40 mL/min/1.73 m^2^) [326]				
Ceftazidime/avibactam	age 66 years (clinical case): 0.94 g every 12 h (CLCR 30–40 mL/min)	100% *fT* > 4 × MIC for ceftazidime 99% *fT* > 4 mg/L for avibactam (MIC = 1.5/4 mg/L)	2.5 g every 8 h (loading dose) 2.5 g every 8 h (maintenance dose)	Concentration in cerebrospinal fluid ≥ 9.4 mg/L—risk of neurotoxicity	[325,331,332]
Ceftobiprole	CLCR < 50 mL/min: 0.5 g as a 2-h intravenous infusion every 12 h	30–40% *T* > MIC MIC = 2 mg/L	NA	NA	[333]
	CLCR < 30 mL/min: 0.25 g as a 2-h intravenous infusion every 12 h				
Carbapenems					
Doripenem	mean age > 60 years, mean CLCR = 53.0 mL/min: 0.5 g every 8 h [258]	40% *fT* > MIC (MIC = 2 μg/mL) [258]	NA	NA	[258]
Ertapenem	mean age 73.1 ± 4.8 years: 1 g every 24 h (normal renal function)	*AUC0-24* 746.1 ± 79.4 μg·h/mL at 1 day *AUC0-24* 681.9 ± 47.0 μg·h/mL at 7 day	1 g every 12 h (loading dose) 1 g every 12 h (maintenance dose)	Plasma concentration > 79.2 µg/mL—risk of neurotoxicity	[325,334,335]
Meropenem	mean age > 65 years, CLCR ≤ 50 mL/min: 1 g every 8 h;	40% *fT*> MIC (MIC≤ 2–8 mg/L)	2 g every 8 h (loading dose) 1 g every 8 h (maintenance dose)	Plasma concentration ≥ 64.2 μg/mL—risk of neurotoxicity *Cmin* ≥ 44.45 μg/mL—risk of nephrotoxicity	[259,336]
	mean age > 65 years, CLCR > 100 mL/min: 2 g every 8 h	40% *T* > MIC (MIC > 8 mg/L)			
Biapenem	mean age > 65 years: 0.3 g every 8 h	40% *T* > MIC (MIC = 2 μg/mL)	NA	NA	[337]

ALT—Alanine transaminase; AST—Aspartate transaminase; *AUC0-24*—Area under the plasma concentration-time curve over the last 24-h dosing interval; CLCR—Creatinine clearance; *Cmin*—Minimum concentration; eGFRcys—Glomerular filtration rate estimated from cystatin C; MIC—Minimum inhibitory concentration; %*T* > MIC—Percent of time for total drug concentration remains above the minimum inhibitory concentration; *fT* > MIC—Percent of time for free drug concentration remains above the minimum inhibitory concentration; AUC—Area under curve; *fCss* > MIC—Free plasma steady-state concentration above the MIC.

**Table 15 biomedicines-11-01633-t015:** Aminoglycosides dosing depending on age, renal function, and hepatic impairment.

Drug	Regimen for Patients with Different Renal Function	PK/PD Target in the Elderly	Regimen for Patients with Hepatic Impairment	Safety	References
Amikacin	mean age > 70 years: 1.8 g every 72 h (CLCR = 40–50 mL/min) 1.8 g every 48 h (CLCR = 60–90 mL/min)	*Cmax* > MIC (MIC ≤ 8 mg/L)	NA	*Cmin* > 4 μg/mL—risk of nephrotoxicity	[312]
Gentamicin	Geriatric population, CLCR > 60 mL/min: 3 mg/kg every 24 h	*Cmax* > MIC (MIC = 1 μg/mL)	NA	*Cmin* > 2 μg/mL—risk of nephrotoxicity	[338]

CLCR—Creatinine clearance; *Cmin*—Minimum concentration; *Cmax*—Maximum concentration; MIC—Minimum inhibitory concentration.

**Table 16 biomedicines-11-01633-t016:** Glycopeptides and Lipopeptides dosing depending on age, renal function, and hepatic impairment.

Drug	Regimen for Patients with Different Renal Function	PK/PD Target in the Elderly	Regimen for Patients with Hepatic Impairment	Safety	References
Glycopeptides					
Vancomycin	mean age ≥ 65 years: 1.0 g every 8 (CLCR > 50 mL/min) 1.0 g every 12 h (CLCR ≤ 50 mL/min)	*Cmin*, ss > MIC	NA	*Cmin* > 20 mg/L—risk of nephrotoxicity	[290]
Lipopeptides					
Daptomycin	eGFRcys = 20 mL/min: age 65 years: 600 mg (loading dose) 350 mg (maintenance dose) every 24 h age 75 years: 550 mg (loading dose) 300 mg (maintenance dose) every 24 h age 85 years: 500 mg (loading dose) 250 mg (maintenance dose) every 24 h age 95 years: 450 mg (loading dose) 200 mg (maintenance dose) every 24 h	*(fAUCss)/*MIC ≥ 66.6	NA	Risk of toxic reactions at *Cmin* > 24 mg/L and *Cmax* > 60 mg/L	[339,340]

CLCR—Creatinine clearance; *Cmin*—Minimum concentration; *Cmin*, ss—Minimum plasma concentration at up to 24 h after administration; *Cmax*—Maximum concentration; eGFRcys—Glomerular filtration rate estimated from cystatin C; MIC—Minimum inhibitory concentration; (*fAUCss*)/MIC—Ratio of the area under the unbound concentration from 0 to 24 h at steady state time curve to the MIC.

**Table 17 biomedicines-11-01633-t017:** Fluoroquinolones dosing depending on age, renal function, and hepatic impairment.

Drug	Regimen for Patients with Different Renal Function	PK/PD Target in the Elderly	Regimen for Patients with Hepatic Impairment	Safety	References
Levofloxacin	mean age 81 years: CLCR 0–19 mL/min: 125 mg every 48 h (MIC = 0.125 mg/L) 250 mg every 48 h (MIC = 0.25 mg/L) 500 mg every 48 h (MIC = 0.5 mg/L) CLCR 20–39 mL/min: 500 mg every 48 h (MIC = 0.125 mg/L) 500 mg every 48 h (MIC = 0.25 mg/L) 750 mg every 48 h (MIC = 0.5 mg/L) CLCR 40–59 mL/min: 500 mg every 48 h (MIC = 0.125 mg/L) 500 mg every 48 h (MIC = 0.25 mg/L) 500 mg every 24 h (MIC = 0.5 mg/L) CLCR 60–79 mL/min: 500 mg every 48 h (MIC = 0.125 mg/L) 750 mg every 48 h (MIC = 0.25 mg/L) 750 mg every 24 h (MIC = 0.5 mg/L) CLCR > 80 mL/min: 750 mg every 48 h (MIC = 0.125 mg/L) 750 mg every 24 h (MIC = 0.25 mg/L) 500 mg every 12 h (MIC = 0.5 mg/L)	*AUC0-24*/MIC ratio (≥95.7)	NA	NA	[261]
Moxifloxacin	No age adjustment 400 mg every 24 h per os	*AUC0-24ss* 46.67 µg·h/mL	NA	NA	[341]

CLCR—Creatinine clearance; MIC—Minimum inhibitory concentration; *AUC0-24*/MIC—Ratio of area under the concentration-time curve during a 24-h period to minimum inhibitory concentration; *AUC0-24ss*—Area under the baseline-corrected plasma concentration versus time curve from time 0 to 24 h at steady state.

**Table 18 biomedicines-11-01633-t018:** Linezolid and polymyxin B dosing depending on age, renal function, and hepatic impairment.

Drug	Regimen for Patients with Different Renal Function	PK/PD Target in the Elderly	Regimen for Patients with Hepatic Impairment	Safety	References
Tedizolid	No age adjustment 200 mg every 24 h	*fAUC*/MIC (MIC ≤0.5 μg/mL)	NA	NA	[303]
Polymyxin B	Median age 68 years (IQR: 63–73), median CRCL 89 (IQR: 68–106) mL/min, bloodstream infection caused by carbapenem-resistant Klebsiella pneumoniae: 1.25 mg/kg every 12 h	*AUC0-24ss*/MIC ≥ 54.4	NA	Risks of nephrotoxicity (manifesations may vary from proteinuria to acute kidney injury) and neurotoxicity	[319,342]

*fAUC*/MIC—The ratio of the area under the bound (unbound) concentration time curve to the MIC; MIC—Minimum inhibitory concentration; *AUC0-24ss*/MIC—Area under the baseline-corrected plasma concentration versus time curve from time 0 to 24 h at steady state.

## Data Availability

Sources of information used in this review are listed in the References.

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
