# Peer review of "Pharmacokinetics of Antibacterial Agents in the Elderly: The Body of Evidence"

_biomedicines, 2023, doi:10.3390/biomedicines11061633_

Round 1

Reviewer 1 Report

The Manuscript Title “Pharmacokinetics of antibacterial agents in the elderly: the 2 body of evidence” has been written well. However, few things need to modify, to add clarity:

Table 7: Add separate header to data title and Add caption to express units in the table. This will help in better visualization and understanding of table.

Add separate column for references in all tables. It will be better to divide tables into multiple tables based on Drug Class.

Author Response

Point 1.

Table 7: Add separate header to data title and Add caption to express units in the table. This will help in better visualization and understanding of table.

Thank you for, we have applied all the recommendations and rearranged Table 7. 

Point 2. Add separate column for references in all tables. It will be better to divide tables into multiple tables based on Drug Class.

Thank you, we have added separate column with References in all the Tables and we have divided Tables 7 and Tables 8 into the multiple Tables based on Drug Class of AB.

Changed manuscript has been submitted.

Reviewer 2 Report

1.     In the abstract part I advise the authors to shorten the background of this part.

2.     Add to the abstract part after the study objective the used methods you follow for your study (the sources of data collection) also the period for this study for example the data collected from 2017-2023).

3.     Add to the abstract part the outcomes of your review and the conclusions including your own future recommendations.

4.     Add after the introduction part the used method of data collection of your review.

5.     I suggest authors to add a section about the antimicrobial resisitence from which the whole world heath systems suffering from especially in elderly patients.

6.     The whole manuscript needs major grammar, typo and editing corrections by a native speakers specialist in biomedical sciences.

Thank you

Author Response

Point 1: In the abstract part I advise the authors to shorten the background of this part.

Response 1: Thank you, we have shorten background in the part I.

Point 2: Add to the abstract part after the study objective the used methods you follow for your study (the sources of data collection) also the period for this study for example the data collected from 2017-2023).

Response 2: Thank you, we have added information in the abstract 1 after the objective. But we want to highlight, that our review article is not a systemic review, that is why implementation of PRISMA principles, along with a a structure of materials, methods, statictical analysis and results can not be applied. We’ll be looking to make a systemic review in the nearest future as a different separate work including the concrete AB Class.

Point 3:    Add to the abstract part the outcomes of your review and the conclusions including your own future recommendations.

Response 3: Thank you, we have added conclusions based on your recommendations. Adding a separate abstract with results is not applicable, since our review article is not a systemic review, but only a literature review, and all the results are considred in the main body.

Point 4:    Add after the introduction part the used method of data collection of your review.

Response 4: We want to highlight, that our review article is not a systemic review, that is why implementation of PRISMA principles, along with a a structure of materials, methods, statictical analysis and results can not be applied. We’ll be looking to make a systemic review in the nearest future as a different separate work including the concrete AB Class

Point 5:   I suggest authors to add a section about the antimicrobial resisitence from which the whole world heath systems suffering from especially in elderly patients.

Response 5: Thank you for sugestion, antibacterial resistance is a dramatically important issue for the elderly, and its decsription will be dependent on the site of infection and pathogens structure, so the perfect presentation of this item will demand about 1 month of work, and we are limited by only 10 days to response. Adding a section of AB resistance will mean adding about 300 new references, (now we have 345), so such inclusion will result in exteremal extension of the publication above the article limits. So, we can plan AB resistance in the elderly as a separate article.

Point 6:   The whole manuscript needs major grammar, typo and editing corrections by a native speakers specialist in biomedical sciences.

Response 6: Thank you for checking the manuscript, we have eliminated grammar, typo and editing corrections by native speaker specialist.

Reviewer 3 Report

Dear Authors, I have read with interest the manuscript and even if the topic could be of interest, I think that the manuscript must be completely rewritten.

I have check the author publications and even if they are pharmacologists only Chenkurov MS have some publications on antimicrobial drugs (TDM), while the others have only review on this topic. I think that they have not a strong skill to write a review on antimicrobial drugs in elderly.

   However I send you my comments:

1) Figure 1 must be revised you need to explain which are the clinical condition that increase the Vd and which reduce it. 

2) The review is on antimicrobial, so general description must be reduced to 1 page.

3) Text please start with the effect  on pharmacokinetic , then pharmacodynamic, then food effects.

3) Please why you describe the effects of Azithromycin and not of Clarithromycin, why you report ampicillin and not amoxicillin, and this is for all antimicrobial drugs    

4) Please delete animal data, these are not relevant for elderly patients

5)  Table 7 must be changed the age must be erased, data of ciprofloxacin are missing

6) In tables you need to describe always the same antimicrobials

7) Section 3.3 the authors describe general data on sarcopenia that are not consistent with common elderly patients and also are not related to antimicrobial drugs but this condition is common in particular patients. So you need to report this in a different section (frail patients). Please report data on antimicrobial drugs, which is the effects of this condition? the dose of each drug must be increased or reduced? or you need to change the time of administration? 

8) Section 4: must be rewritten considering the clinical data. How clinicians need to change the dosage of each drug reported? what do you think with "Risk of transient low level elevations of ALT or AST in serum"? what is this clinical effects? what do you think with frail patients? you did not report this in the text and in other tables. MIC is for the microrganism and not for antimicrobials so when you report it you need to decribe for which microrganism it is described.

  9) I have not read a pharmacological differentiation between the administration of drugs considering their time-dependent or concentration-dependent effects

10) Add a table considering the differentiaton of drug regarding their lipophilic and hydrophilic characteristics

11) please add a section on drug interactions 

None

Author Response

Response to Reviewer 3 Comments

General comment: I have check the author publications and even if they are pharmacologists only Chenkurov MS have some publications on antimicrobial drugs (TDM), while the others have only review on this topic. I think that they have not a strong skill to write a review on antimicrobial drugs in elderly.

Response to the general comment: Dear reviewer 3, we have added a list with examples of our published articles dedicated to AB, their PK and other related studies. Please, note the authorship of the given works (last 3 years only):

  1. Butranova OI, Ushkalova EA, Zyryanov SK, Chenkurov MS. Developmental Pharmacokinetics of Antibiotics Used in Neonatal ICU: Focus on Preterm Infants. Biomedicines. 2023; 11(3):940. https://doi.org/10.3390/biomedicines11030940
  2. Zyryanov S, Bondareva I, Butranova O, Kazanova A (2023), Population PK/PD modelling of meropenem in preterm newborns based on therapeutic drug monitoring data. Front. Pharmacol. 14:1079680. DOI: 10.3389 fphar.2023.1079680
  3. Zyryanov SK, Ushkalova EA, Kondratyeva EI, Butranova OI, Kondakova YA. Gene Polymorphism of Biotransformation Enzymes and Ciprofloxacin Pharmacokinetics in Pediatric Patients with Cystic Fibrosis. Biomedicines. 2022;10(5):1050. DOI:10.3390/biomedicines10051050. 
  4. Kuzmina, A.V., Asetskaya, I.L., Zyryanov, S.K. et al. Detecting medication errors associated with the use of beta-lactams in the Russian Pharmacovigilance database. BMC Pharmacol Toxicol 22, 5 (2021). https://doi.org/10.1186/s40360-020-00470-x
  5. Zyryanov S.K., Butranova O.I. Treatment for resistant vulvovaginal candidiasis. – Obstetrics and gynecology. - 2022. – № 1. – p. 136-146. – DOI 10.18565/aig.2022.1.136-146.
  6. Beloborodov V.B., Goloschapov O.V., Gusarov V.G., Dekhnich А.V., Zamyatin M.N., Zubareva N.A., Zyryanov S.K., Kamyshova D.A., Klimko N.N., Kozlov R.S., Kulabukhov V.V., Petrushin M.A., Polushin Yu.S., Popov D.A., Rudnov V.A., Sidorenko S.V., Sokolov D.V., Shlyk I.V., Edelshtein M.V., Yakovlev S.V. Guidelines of the Association of Anesthesiologists-Intensivists, the Interregional Non-Governmental Organization Alliance of Clinical Chemotherapists and Microbiologists, the Interregional Association for Clinical Microbiology and Antimicrobial Chemotherapy (IACMAC), and NGO Russian Sepsis Forum "Diagnostics and antimicrobial therapy of the infections caused by multiresistant microorganisms" (update 2022). Messenger of ANESTHESIOLOGY AND RESUSCITATION. 2022;19(2):84-114. (In Russ.) https://doi.org/10.21292/2078-5658-2022-19-2-84-114
  7. Baranov A.A., Namazova-Baranova L.S., Kutsev S.I., Avdeev S.N., Polevichenko E.V., Belevskiy A.S., Kondratyeva E.I., Simonova O.I., Kashirskaya N.Yu., Sherman V.D., Voronkova A.Yu., Amelina E.L., Gembitskaya T.E., Krasovskiy S.A., Chermenskiy A.G., Stepanenko T.A., Selimzyanova L.R., Vishneva E.A., Gorinova Yu.V., Roslavtseva E.A., Asherova I.K., Ilyenkova N.A., Zyryanov S.K., Odinayeva N.D., Maksimycheva T.Yu., Orlov A.V., Semykin S.Yu., Chernukha M.Yu., Shaginyan I.A., Avetisyan L.R., Shumkova G.L., Krylova N.A., Dronov I.A., Kostyleva M.N., Zhelenina L.A., Klimko N.N., Borzova Yu.V., Vasilyeva N.V., Bogomolova T.S., Speranskaya A.A., Baranova I.A., Furman E.G., Shadrina V.V., Shchapov N.F., Petrova N.V., Pashkov I.V., Tsirulnikova O.M., Polyakov D.P., Svistushkin V.M., Sin'kov E.V., Chernykh V.B., Repina S.A., Blagovidov D.A., Kostinov M.P., Kondratenko O.V., Lyamin A.V., Polikarpova S.V., Polyakov A.V., Adyan T.A., Goldshtein D.V., Bukharova T.B., Efremova A.S., Ovsyankina E.S., Panova L.V., Cherkashina I.V. Modern Approaches in Management of Children with Cystic fibrosis. Pediatric pharmacology. 2022;19(2):153-195. (In Russ.) https://doi.org/10.15690/pf.v19i2.2417
  8. 8. Gostev V, Ivanova K, Kruglov A, Kalinogorskaya O, Ryabchenko I, Zyryanov S, Kalisnikova E, Likholetova D, Lobzin Y, Sidorenko S. Comparative genome analysis of global and Russian strains of community-acquired methicillin-resistant Staphylococcus aureus ST22, a ‘Gaza clone’. Int J Antimicrob Agents. 2021 Feb;57(2):106264. DOI: 10.1016/j.ijantimicag.2020.106264. Epub 2020 Dec 14. PMID: 33326849.
  9. Zyryanov S.K., Baybulatova E.A. Clinical and pharmacological analysis of antiseptics used in practical medicine. Vopr. prakt. pediatr. (Clinical Practice in Pediatrics). 2021; 16(6): 77–92. (In Russian). DOI: 10.20953/1817-7646-2021-6-77-92
  10. Bondareva I.B., Zyryanov S.K., Kazanova A.M. Population Pharmacokinetics of Meropenem in Preterm Infants // Annals of the Russian academy of medical sciences. - 2021. - Vol. 76. - N. 5. - P. 497-505. doi: 10.15690/vramn1449
  11. Logunov DY, Dolzhikova IV, Shcheblyakov DV, Zyryanov SK et al. Safety and efficacy of an rAd26 and rAd5 vector-based heterologous prime-boost COVID-19 vaccine: an interim analysis of a randomised controlled phase 3 trial in Russia [published correction appears in Lancet. 2021 Feb 20;397(10275):670]. Lancet. 2021;397(10275):671-681. doi:10.1016/S0140-6736(21)00234-8
  12. Zyryanov S.K., Butranova O.I., Chenkurov M.S. Chloramphenicol: new possibilities of the old drug // Obstetrics and gynecology. – 2021. – № 11. – p. 81-94. – DOI 10.18565/aig.2021.11.81-94.
  13. Bondareva I.B., Zyryanov S.K., Chenkurov M.S. Pharmacokinetic analysis of meropenem therapeutic drug monitoring data (TDM) in critically ill adult patients. // Antibiotics and chemotherapy. – 2021. – Vol. 66, № 11-12. – p. 31-38. – DOI 10.37489/0235-2990-2021-66-11-12-31-38.
  14. Zyryanov S.K., Butranova O.I., Al-Ragawi A. Modern Pharmacotherapy of uncomplicated infections of the lower urinary tract: the position of antibacterial and herbal preparations // Urologiia. – 2020. – № 2. – p. 76-84. – DOI 10.18565/urology.2020.2.76-84. – EDN GZQXOT.
  15. Zyryanov S.K., Baybulatova E.A. Rifaximin-Alpha and Other Crystalline Forms of Rifaximin: Are There Any Differences? Antibiotics and Chemotherapy. 2020;65(7-8):52-62. (In Russ.) https://doi.org/10.37489/0235-2990-2020-65-7-8-52-62
  16. Zyryanov S.K., Baybulatova E.A. The Use of New Dosage Forms of Antibiotics as a Way to Improve the Effectiveness and Safety of Antibiotic Therapy. Antibiotics and Chemotherapy. 2019;64(3-4):81-91. (In Russ.) – DOI: 10.24411/0235-2990-2019-10020

Point 1: Figure 1 must be revised you need to explain which are the clinical condition that increase the Vd and which reduce it.

Response 1: Thank you, we have revised Fig. 1 and added explanations.

Point 2: The review is on antimicrobial, so general description must be reduced to 1 page.

Response 2: Thank you, we have shorten the general description, it is less than 1 page now.

Point 3:  Text please start with the effect  on pharmacokinetic , then pharmacodynamic, then food effects.

Response 3: Thank you, the text is dedicated to the review of pharmacokinetics, so each parameter is described following ADME.

Point 4:   Please why you describe the effects of Azithromycin and not of Clarithromycin, why you report ampicillin and not amoxicillin, and this is for all antimicrobial drugs   

Response 4: Thank you, we have described ABs based on the available published data and on the principle of the relative nature of the drugs according to their structures and similarities of PK profile, revealing the most important issues for the elderly.

Point 5:  Please delete animal data, these are not relevant for elderly patients

Response 5: Thank you, we have deleted animal data from the Tables highlighting features in the elderly.

Point 6:   Table 7 must be changed the age must be erased, data of ciprofloxacin are missing

Response 6: Thank you, we have rearranged the Table 7, though we want to pay attention, that age indication in the elderly is important, since they are a highly heterogenous population, and profile in 65 yo patient may be drammatically different from those 75 or 85 yo, as it is revealed in the results.

Point 7:   In tables you need to describe always the same antimicrobials

Response 7: Thank you for recommendation, our aim was to perform the most relevant and full information about the PK of AB in the elderly, and available publications not always highlifgt the same antibiotics regarding certain PK parameters discussed in the review.

Point 8:   Section 3.3 the authors describe general data on sarcopenia that are not consistent with common elderly patients and also are not related to antimicrobial drugs but this condition is common in particular patients. So you need to report this in a different section (frail patients). Please report data on antimicrobial drugs, which is the effects of this condition? the dose of each drug must be increased or reduced? or you need to change the time of administration?

Response 8: Thank you for the comment. In the section 3.3 we have indicated sarcopenia only according to the results of the study by Kaburaki S et al (2022). Text is “Changes in the body composition specific for the elderly may affect functions of drug metabolizing enzymes. Kaburaki S et al (2022) observed associations between the skeletal muscle mass index (SMI), handgrip strength (HGS), hepatic steatosis index, and activity of CYP2C19 and CYP3A4. In male patients ≥65 years of age a reduction in SMI and HGS below the sarcopenia diagnostic criteria correlated with a decline in CYP2C19 and CYP3A4 activity. In elderly female patients, a decline in CYP2C19 met-abolic activity was associated with fatty liver disease presence [162].”

Adding a separate section on frailty will mean general description of the geriatric syndrome, bur our review is dedicated to PK of AB. Certain effects of frailty on PK of Abs are given in the sections dedicated to Distribution and Metabolism.

Point 9. Section 4: must be rewritten considering the clinical data. How clinicians need to change the dosage of each drug reported? what do you think with "Risk of transient low level elevations of ALT or AST in serum"? what is this clinical effects? what do you think with frail patients? you did not report this in the text and in other tables. MIC is for the microrganism and not for antimicrobials so when you report it you need to decribe for which microrganism it is described.

Response 9: Thank you for the comment. In the section 4 we were using only clinical data, and summary Table contains information about the recommendations how to change the dosage for each drug.

"Risk of transient low level elevations of ALT or AST in serum" – it means, that use of a given AB may contribute to the transient damage to the liver, as it is considred in the clinical practice. Additional explanation for non-clinicians has been added to the table.

MICs were given based on the used clinical trials according to the references. According to the practice, the given variant may reflect the response of the majority of typical pathogens, of course  exept MDR microflora.

Point 10. I have not read a pharmacological differentiation between the administration of drugs considering their time-dependent or concentration-dependent effects

Response 10: Thank yu for the comment, the differenceis mainly in the sphere of pharmacodynamics, so PK/PD indices indicated in the last table in the section 4 are the main markers of differentiation.

Point 11: Add a table considering the differentiaton of drug regarding their lipophilic and hydrophilic characteristics

Response 11: Thank you for comment. There are multiple articles published including tables with such comparison. In our work we have indicated in the text lipo- or hydrophilicity of the main AB. Their properties are indicated in the tables describing certain PK parameters, that is why this information is presented in our work.

Point 12: Please add a section on drug interactions.

Response 12: Thank you for recommendation, it is really an interesting issue. Main drug interactions are seen on the stage of absorption and drug metabolism. In our work we have highlighted the main vulnerable points in this regard. Preparing a separate section in a perfect way will extent the article content twicely, so we’ll be planning to perform this issue in a separate article.

Round 2

Reviewer 1 Report

Thanks for making changes. Manuscript is acceptable in current form.

Author Response

Dear reviewer 1!

We are grateful for the work done in reviewing our article!

Reviewer 2 Report

The authors conducted all the required corrections and I have not any more comments

Author Response

(The authors gave the same response as above.)

Reviewer 3 Report

Dear Authors,

I have read again the manuscript and I think that even if it has been revised in some sections, it has not be changed as requested.

It seems to be a revision of other publications and it is not different respect to these. In fact you write "we have indicated sarcopenia only according to the results of the study by Kaburaki S et al (2022)"

I think that a review on antimicrobials in elderly is very important and readers must be know what is the difference in the dosage and in the time of administration. This review even if is very well written does not give these informations that are requested in the first review. 

The definition increased or reduced is very general this increase, what is the amount of increase or decrease, this is important for a new review of elderly patients. Moreover as requested in the previous review I have not read particular data that can differentiate adult and elderly but only generic data on antimicrobial drugs without indications on the very important problem in the elderly: renal failure, liver failure, poly-therapy  

Author Response

Dear reviewer 3!

Thank you for the revision of our work!

Point 1. I have read again the manuscript and I think that even if it has been revised in some sections, it has not be changed as requested.

Response 1:  Thank you, we have added all the corrections to the Figure, Tables, main sections, added information regarding your recommendations. The only position we discussed was adding a section about drug interactions. It is extremely important, but it is a theme for a single article since the volume of exact data describing each group of antibacterials is extra huge.

Point 2. It seems to be a revision of other publications and it is not different respect to these. In fact you write "we have indicated sarcopenia only according to the results of the study by Kaburaki S et al (2022)"

Response 2: Thank you, yes, our work s a review of published materials involving PK of ABs in the elderly, search has been made in the database Pubmed from 1980-s to 2023. The effect of sarcopenia on PK of ABs was described through several sections, regarding the change of body composition.

Point 3. I think that a review on antimicrobials in elderly is very important and readers must be know what is the difference in the dosage and in the time of administration. This review even if is very well written does not give these informations that are requested in the first review.

Response 3: Thank you, information about the difference in the dosage and in the time of administration of ABs in the elderly (including renal failure and hepatic insufficiency) has been extensively described in the Tables located in the last section, section 4, “AB Dosing Regimens in the Elderly”

Point 4. The definition increased or reduced is very general this increase, what is the amount of increase or decrease, this is important for a new review of elderly patients. Moreover as requested in the previous review I have not read particular data that can differentiate adult and elderly but only generic data on antimicrobial drugs without indications on the very important problem in the elderly: renal failure, liver failure, poly-therapy 

Response 4: The definitions “increased or reduced” regarding ALT and AST are standard definitions in the clinical practice, which are used to indicate appearance of a damage to hepatocytes, though this damage is typically reversable. It means possibility of a hepatotoxic reaction produced by a drug.  Regarding renal failure, liver failure, polypharmacy – these problems were discussed in each subsection dedicated to the concrete pharmacokinetic parameters of ABs.   
